# PAF1C restores transcription after DNA damage independently of promoting histone mark deposition

Janne J M van Schie, Bram A F J de Groot [ID], Diana van den Heuvel [ID] & Martijn S Luijsterburg [ID] ✉

## Abstract

When RNA polymerase II (RNAPII) stalls at transcription-blocking lesions, the transcription-coupled DNA repair (TCR) pathway is activated to remove the damage. After repair, efficient transcription restart requires the PAF1 elongation complex (PAF1C). PAF1C promotes deposition of transcription-associated histone marks, which are enriched at active genes and proposed to support post-repair transcription recovery. Using conditional knockouts of writers of PAF1C-associated histone marks, we show that deposition of $H3K79_{me}$, $H3K4_{me3}$, and $H2BK120_{Ub}$ is dispensable for transcription restart. While $H3K4_{me3}$ levels remain mostly unchanged during DNA damage-induced transcription inhibition, $H2BK120_{Ub}$ levels decrease after damage and are co-transcriptionally restored upon transcription restart. We further find that, unlike the core subunits PAF1 and CTR9, the dissociable PAF1C subunit RTF1 does not contribute to transcription restart. Finally, we show that the TCR core factor CSB is not required for the transient PAF1C interaction during normal transcription, but specifically stabilizes PAF1C on RNAPII in response to DNA damage-induced stalling. Together, these findings indicate that transcription restoration after DNA damage is driven by PAF1C independently of transcription-associated histone mark deposition.

Subject Categories Chromatin, Transcription & Genomics; DNA Replication, Recombination & Repair

## Introduction

The presence of DNA damage in the transcribed strand presents a major obstacle to transcription, leading to persistent stalling of RNA polymerase II (RNAPII) at bulky DNA lesions. This stalling is highly toxic and triggers a genome-wide transcriptional arrest (Nakazawa et al, 2020). While DNA lesions primarily lead to the direct stalling of elongating RNAPII *in cis* (Moné et al, 2001), a regulated inhibition of transcription initiation in trans via the stress-induced repressor ATF3 and RNAPII degradation has also been proposed (Epanchintsev et al, 2017; Proietti-De-Santis et al,

2006; Rockx et al, 2000; Tufegdzic Vidakovic et al, 2020). Overcoming transcriptional arrest and restoring transcription after DNA repair is essential for maintaining proper gene expression and cellular homeostasis.

To remove transcription-blocking DNA lesions, cells rely on the transcription-coupled DNA repair (TCR) pathway. TCR is initiated by the coordinated recruitment of the TCR-specific proteins CSB, CSA, UVSSA and STK19 to DNA damage-stalled RNAPII assisted by elongation factor ELOF1 (Geijer et al, 2021; Kokic et al, 2021; Kokic et al, 2024; Mevissen et al, 2024; Ramadhin et al, 2024; Tan et al, 2024; van den Heuvel et al, 2024; van den Heuvel et al, 2021; van der Meer and Luijsterburg, 2025; van der Weegen et al, 2021; van der Weegen et al, 2020; van Sluis et al, 2025). Together, these factors promote the recruitment of the TFIIH complex to damage-stalled RNAPII, leading to either backtracking or displacement of RNAPII to facilitate access to the DNA lesion by downstream repair proteins (Gonzalo-Hansen et al, 2024; van den Heuvel et al, 2024). Subsequently, general nucleotide excision repair (NER) proteins, such as XPA, RPA, and the endonucleases XPG and ERCC1-XPF, coordinate dual incision around the lesion to remove the damaged DNA strand, followed by gap fill DNA synthesis and ligation (Marteijn et al, 2014). Mutations in TCR components CSB and CSA cause Cockayne syndrome, a progeroid disorder characterized by severe developmental and neurological dysfunction (Laugel et al, 2010; Lehmann, 2003).

While the removal of DNA lesions by TCR is essential, it is not sufficient for transcription restart. TCR involves displacement of RNAPII from the damaged DNA template (Chiou et al, 2018; Gonzalo-Hansen et al, 2024; van der Meer et al, 2026; Zhu et al, 2024), followed by transcription restart from gene promoters. Post-repair transcription restart may require alleviation of ATF3-mediated transcriptional repression at promoters via CSB- and CSA-mediated degradation of ATF3 (Epanchintsev et al, 2017; Kristensen et al, 2013). In addition, chromatin remodeling has been implicated in post-repair transcription restart (Dinant et al, 2013; Mandemaker et al, 2014). These observations suggest that post-repair transcription pathways are at least partially distinct from those required for general transcription. However, the precise mechanisms underlying transcription recovery and its coordination with TCR remain to be fully elucidated.

The PAF1 subunit of the RNA polymerase II-associated factor complex (PAF1C) has emerged as a critical regulator of transcriptional restart following genotoxic stress (van den Heuvel et al, 2021). PAF1C, composed of the subunits CTR9, PAF1, LEO1, CDC73, SKI8, and the loosely associated subunit RTF1, localizes to

Department of Human Genetics, Leiden University Medical Center, Leiden, The Netherlands. ✉E-mail: m.luijsterburg@lumc.nl

actively transcribed genes through its association with RNAPII to regulate transcription elongation, promoter-proximal pausing, and deposition of histone modifications (Chen et al, 2015a; Chen et al, 2021; Chen et al, 2017; Hou et al, 2019; Yu et al, 2015; Zumer et al, 2021). Which PAF1C functions contribute to transcription restart, and whether subunits other than PAF1 are required, remains unknown.

PAF1C recruits the E2 ubiquitin conjugase RAD6A (also known as UBE2A) and the RING-type E3 ubiquitin ligase complex RNF20-RNF40 to RNAPII to co-transcriptionally mono-ubiquitylate histone H2B at lysine 120 (H2BK120$_{Ub}$) (Fetian et al, 2023; Van Oss et al, 2016). In both yeast and humans, this modification stimulates deposition of H3K4 trimethylation (H3K4$_{me3}$) by SET1/COMPASS complexes and H3K79 methylation (H3K79$_{me}$) by DOT1L (Fetian et al, 2023; Kim et al, 2009; Krogan et al, 2003; Valencia-Sanchez et al, 2019; Wood et al, 2003; Worden et al, 2019). Thus, PAF1C promotes the deposition of H2BK120$_{Ub}$, H3K4$_{me3}$, and H3K79$_{me}$ (Francette et al, 2021), all associated with active transcription (Gates et al, 2017; Zhang et al, 2015). H3K4$_{me3}$ is enriched near promoters and correlates with transcription initiation and promoter-proximal pause release (Hu et al, 2023; Wang et al, 2023), while H2BK120$_{Ub}$ and H3K79$_{me}$ are enriched in gene bodies and have been linked to transcription elongation and the DNA damage response (Jeusset and McManus, 2022; Ljungman et al, 2019; Mattiroli and Penengo, 2021; Werner et al, 2025). These modifications have also been associated with post-repair transcription restart: UV-induced H3K4 methylation facilitates transcriptional reactivation in *C. elegans* (Wang et al, 2020), and DOT1L-mediated H3K79 methylation contributes to transcription restart in mouse cells (Oksenych et al, 2013). Furthermore, H2BK120$_{Ub}$ levels are reduced following UV damage (Mao et al, 2014; van den Heuvel et al, 2021), suggesting a potential interplay between transcription-associated histone modifications and transcription recovery (van den Heuvel et al, 2021), although this has not been directly assessed in human cells.

In this study, we found that the transcription-associated histone modifications H2BK120$_{Ub}$, H3K4$_{me3}$, and H3K79$_{me}$ are dispensable for transcriptional recovery following genotoxic stress. In addition, we find that transcription restart is specifically supported by the PAF1 core complex, including the PAF1 and CTR9 subunits, whereas the dissociable subunit RTF1 is not required. Together, these findings suggest that the restoration of transcription after DNA damage depends on the ability of PAF1C to directly promote processive RNAPII elongation, independently of transcription-associated histone mark deposition.

# Results

## 5-EU incorporation predominantly reflects RNAPII-driven transcription

Recovery of RNA synthesis (RRS), measured by 5-ethynyl uridine (5-EU) incorporation during nascent transcription, is widely used to assess transcription restart following DNA damage (Carnie et al, 2024; Geijer et al, 2021; Kokic et al, 2024; Ramadhin et al, 2024; van den Heuvel et al, 2024; van den Heuvel et al, 2021; van der Meer et al, 2026; van der Weegen et al, 2021; van Sluis et al, 2024). However, the relative contributions of RNAPI, RNAPII, and

RNAPIII to this assay are not well established. To address this, we measured 5-EU incorporation after treating RPE1 cells for 4 h with RNA polymerase–specific inhibitors: BMH-21 (RNAPI), 5,6-dichlorobenzimidazole 1-β-D-ribofuranoside (DRB; RNAPII), and ML-60218 (RNAPIII) (Fig. EV1A). Combined inhibition of RNAPI and RNAPII reduced 5-EU incorporation to background levels, whereas RNAPIII inhibition did not reduce nascent transcription at all. Individual inhibition revealed that RNAPI contributes ~25% of the detected signal, predominantly from highly transcribed regions corresponding to the nucleoli, whereas RNAPII contributes ~75% throughout the nucleoplasm (Fig. EV1B). Thus, RNAPII-driven transcription is the predominant contributor to the 5-EU signal in our RRS assays.

## DOT1L and H3K79 methylation are dispensable for transcription restart following UV

H3K79$_{me}$ is a histone mark enriched in the gene bodies of actively transcribed genes and correlates with high RNAPII elongation rates (Ljungman et al, 2019). However, loss of H3K79$_{me}$ does not appear to affect transcription elongation rates (Cao et al, 2020; Wu et al, 2021) and only causes minimal changes in gene expression (Bernt et al, 2011; Ho et al, 2013). H3K79 methylation has been linked to transcription initiation (Wu et al, 2021), but the precise contribution of H3K79$_{me}$ to transcription remains unclear (Ljungman et al, 2019). No obvious H3K79 demethylase has been identified (Ljungman et al, 2019; Wille and Sridharan, 2022). Instead, reduction of H3K79$_{me}$ seems to rely on histone replacement or dilution through cell division (Farooq et al, 2016). Consequently, H3K79$_{me}$ has a relatively long half-life (multiple days) (Barth and Imhof, 2010), making it unlikely to dynamically respond to transcription-blocking DNA damage. Previous studies have shown that H3K79$_{me}$ facilitates transcription restart after genotoxic stress in mouse cells, possibly by opening chromatin to allow RNAPII reactivation (Oksenych et al, 2013). Since differences in chromatin requirements for TCR between mouse and human cells have been reported previously (Apelt et al, 2020), we tested whether transcription restart after UV irradiation depends on H3K79$_{me}$ in human cells.

To test the role of H3K79$_{me}$ in transcription restart following UV, we inhibited DOT1L, the sole known H3K79 methyltransferase (van Leeuwen et al, 2002), using the inhibitor Pinometostat (DOT1Li) in human RPE1-hTERT cells. We treated cells with DOT1Li for 7 days, which resulted in the loss of H3K79$_{me}$ as detected by western blot analysis using an antibody recognizing H3K79$_{me2}$ (Fig. 1A), and performed RRS assays after UV irradiation. In all conditions, RNA synthesis dropped 3 h after UV exposure. As expected, TCR-deficient CSB$^{KO}$ cells failed to recover RNA synthesis after 24 h. In contrast, nascent RNA synthesis in wild-type (WT) cells treated with DOT1Li recovered to levels similar to those of untreated WT cells (Fig. 1B,C). Similar results were observed in human 48BR-hTERT cells treated with DOT1Li, whereas inhibition with the neddylation inhibitor NAEi blocked transcription recovery (Fig. 1D–F) as expected (Kokic et al, 2024; Nakazawa et al, 2020; Tufegdzic Vidakovic et al, 2020). To address potential incomplete inhibition of H3K79 methylation or H3K79$_{me}$-independent DOT1L functions (Cao et al, 2020), we generated two clonal DOT1L$^{KO}$ cell lines using crRNAs targeting exon 2 and exon 5. DOT1L$^{KO}$ clones were viable and lacked

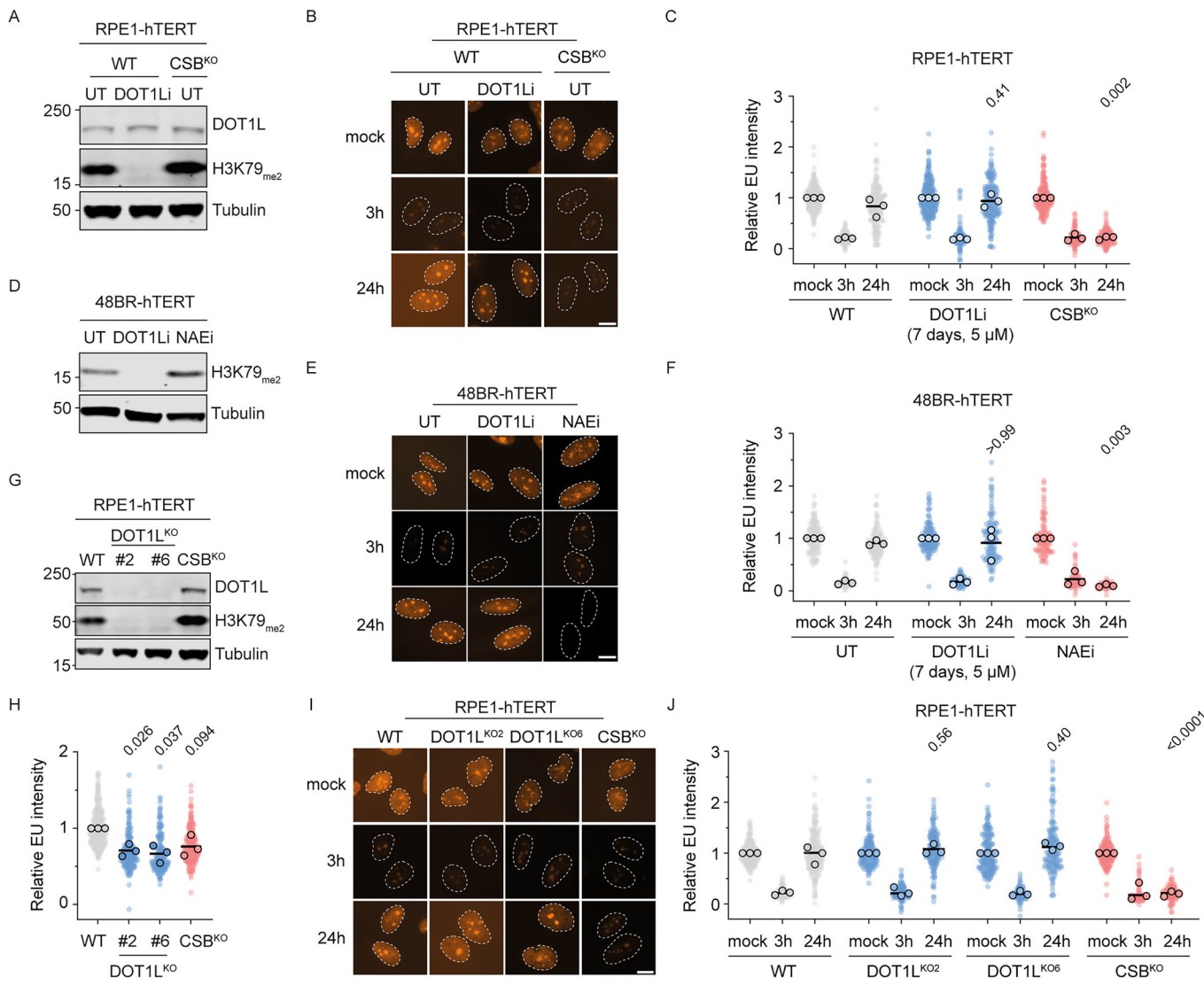

**Figure 1. H3K79me is dispensable for transcription restart after DNA damage.**

(A) Western blot with indicated antibodies of WT and CSB^KO cells untreated (UT) or treated with 5 µM DOT1Li (pinometostat) for 7 days. (B) Representative images of RNA recovery assay (RRS) of conditions from (A). Cells were labeled with 5-ethynyl uridine (EU) for 1 h in unirradiated (mock) condition or 3 h and 24 h after 12 J/m² UV-C. Dashed lines represent the nucleus defined by DAPI staining. Scale bar, 10 µm. (C) Quantification of RRS from conditions in (B). Cells are depicted as individual data points. Circles are individual means of three biological replicates. Lines indicate means of the three biological replicates. Scale bar, 10 µm. Statistical significance was determined by one-way ANOVA on the means of biological replicates with Dunnett's multiple comparisons test against mock 24 h condition. (D) Western blot of 48BR-hTERT cells untreated, treated with 5 µM DOT1Li (pinometostat) for 7 days or treated for 24 h with 10 µM NAEi (MLN4924). (E) Representative images of RRS of conditions from (D). Dashed lines represent the nucleus defined by DAPI staining. Scale bar, 10 µm. (F) Quantification of RRS from (E) as in (C). (G) Western blot of RPE1 cells with the indicated genotype. (H) Nascent RNA synthesis of nonirradiated DOT1L^KO and CSB^KO cells relative to WT cells. Circles are individual means of three biological replicates, lines indicate means of the biological replicates. Statistical significance determined by a one-sample t test with hypothetical value 1. (I) Representative images of RRS of cells from (G). Dashed lines represent the nucleus defined by DAPI staining. Scale bar, 10 µm. (J) Quantification of RRS from (I) as in (C). Source data are available online for this figure.

detectable DOT1L protein or H3K79me (Fig. 1G). We observed slightly but significantly reduced levels of nascent transcription in unirradiated cells (Fig. 1H), possibly reflecting DOT1L's role in promoting elongating RNAPII levels on chromatin (Wu et al, 2021). Similar to DOT1Li-treated cells, DOT1L^KO cells recovered nascent transcription comparable to WT cells 24 h after UV irradiation (Fig. 1I,J). These observations indicate that DOT1L and H3K79 methylation are dispensable for transcription restart following genotoxic stress in human cells.

## The SET1 complex and H3K4 trimethylation are dispensable for transcriptional restart

H3K4me3 is enriched near promoters and is believed to facilitate transcription initiation and the release of promoter-proximal paused RNAPII (Hu et al, 2023; Wang et al, 2023). Consequently, RNAPII restart from promoters following successful repair of transcription-blocking DNA damage may depend on H3K4me3. Deposition of H3K4me3 is mediated by six SET1-like histone

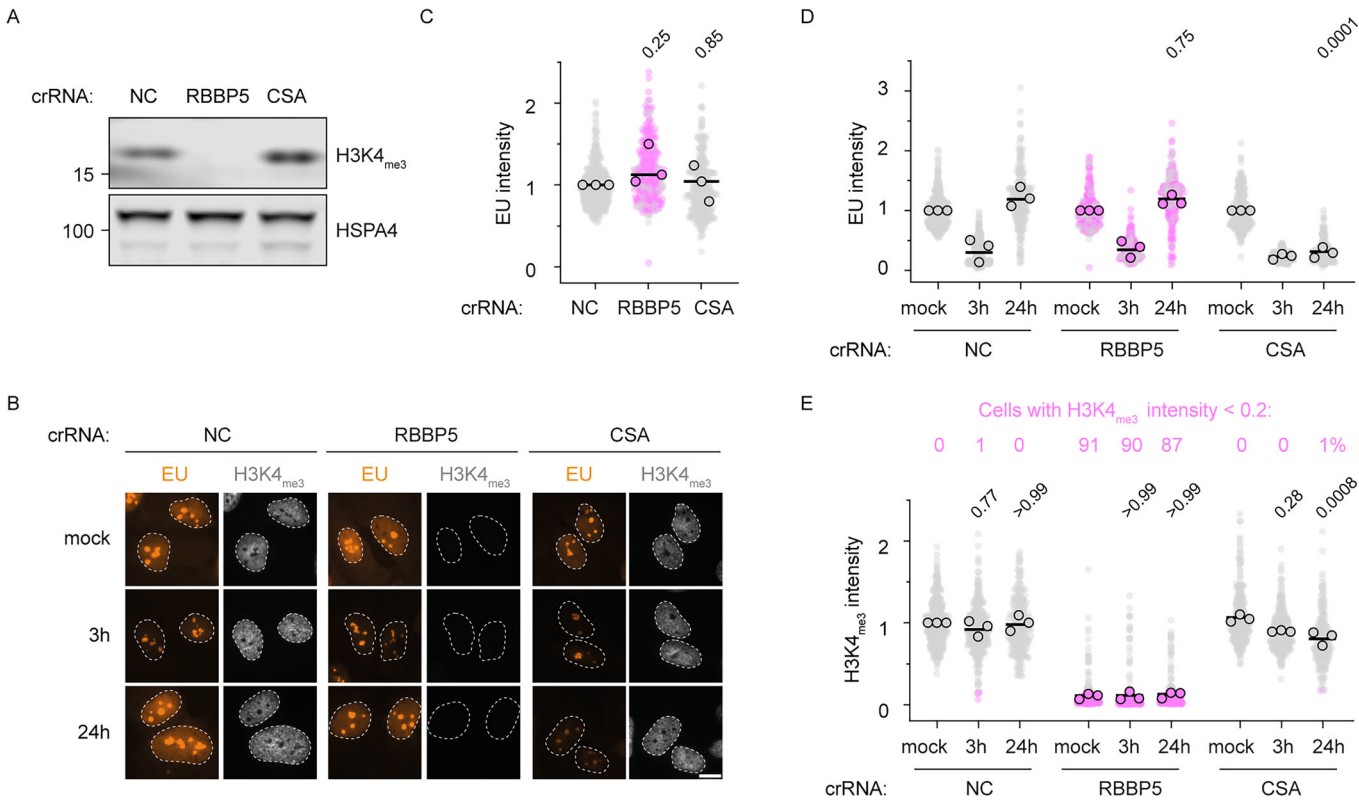

**Figure 2. H3K4me₃ is dispensable for transcription restart after DNA damage.**

(**A**) Western blot of RPE1 cells 5 days after Cas9 induction and transfection with indicated crRNAs. As a negative control (NC) a crRNA targeting olfactory receptor 10A7 (OR10A7) was used. (**B**) Representative images of cells labeled for 1 h with EU, unirradiated (mock), 3 h and 24 h after 12 J/m² UV-C, 5 days after transfection with indicated crRNAs. Cells were co-stained for H3K4me₃. Dashed lines represent the nucleus defined by DAPI staining. Scale bar, 10 μm. (**C**) Nascent EU incorporation in nonirradiated cells transfected with indicated crRNAs relative to control-transfected cells (NC). Statistical significance on the means of three biological replicates was determined by a one-sample *t* test with hypothetical value 1. (**D**) Quantification of EU levels in conditions from (**B**), normalized to the mock condition per crRNA. Statistical significance was determined by ordinary one-way ANOVA on the means of three biological replicates with Dunnett's multiple comparisons test against crNC 24 h condition. (**E**) Quantification of H3K4me₃ levels in conditions from (**B**), normalized to the NC mock condition. Individual data points represent cells, circles represent means from biological replicates. Pink dots represent cells with relative H3K4me₃ intensity levels below 0.2, numbers indicate the percentage of cells with relative H3K4me₃ levels below 0.2. Statistical significance was determined by one-way ANOVA on the means of three biological replicates with Dunnett's multiple comparisons test against the respective mock condition. Source data are available online for this figure.

methyltransferase complexes, whose H3K4 methyltransferase activities all depend on the shared WRAD complex, composed of the core subunits WDR5, RBBP5, ASH2L, and DPY30 (Jiang, 2020). To test the involvement of H3K4me₃ in transcription restart after UV, we depleted the shared core subunit RBBP5. Because SET1 complexes are essential for cell viability, likely due to their central role in transcriptional regulation (Jiang, 2020), we were unable to generate viable RBBP5KO clones. Instead, we performed conditional knockout of RBBP5 using CRISPR/Cas9 and conducted assays after 5 days, when cells were still viable but exhibited very low H3K4me₃ levels (Fig. 2A).

We then measured nascent transcription by 5-EU incorporation while co-staining for H3K4me₃ (Fig. 2B), which confirmed loss of H3K4me₃ in crRBBP5-transfected cells. Despite near-complete loss of H3K4me₃, nascent transcription levels were comparable to control cells (Fig. 2C). RRS assays with H3K4me₃ co-staining (Fig. 2B,D,E) showed that while RBBP5-depleted cells had H3K4me₃ levels <20% of that of WT cells in about 90% of the cells (pink dots in Fig. 2C–E), they recovered transcription like WT cells. H3K4me₃

levels were not reduced at 3 h after UV in cells transfected with negative control crRNAs, even when nascent transcription was reduced by ~80% (Fig. 2B,D,E). CSA-depleted cells at 3 and 24 h after UV exposure showed slightly reduced, yet significantly lower, levels of H3K4me₃, possibly reflecting the contribution of decreased transcription to H3K4me₃ levels in these cells. Together, these results demonstrate that the WRAD complex and H3K4me₃ are not required for the immediate restart of transcription following the repair of transcription-blocking DNA damage.

## H2BK120 ubiquitylation and RNF20 are not essential for transcription restart

H2BK120Ub is a co-transcriptionally deposited histone mark enriched within the gene bodies of actively transcribed genes (Fetian et al, 2023) and has been ascribed multiple functions in the DNA damage response (Mattiroli and Penengo, 2021). The levels of H2BK120Ub decrease following UV-induced DNA damage (Mao et al, 2014; van den Heuvel et al, 2021), suggesting a potential

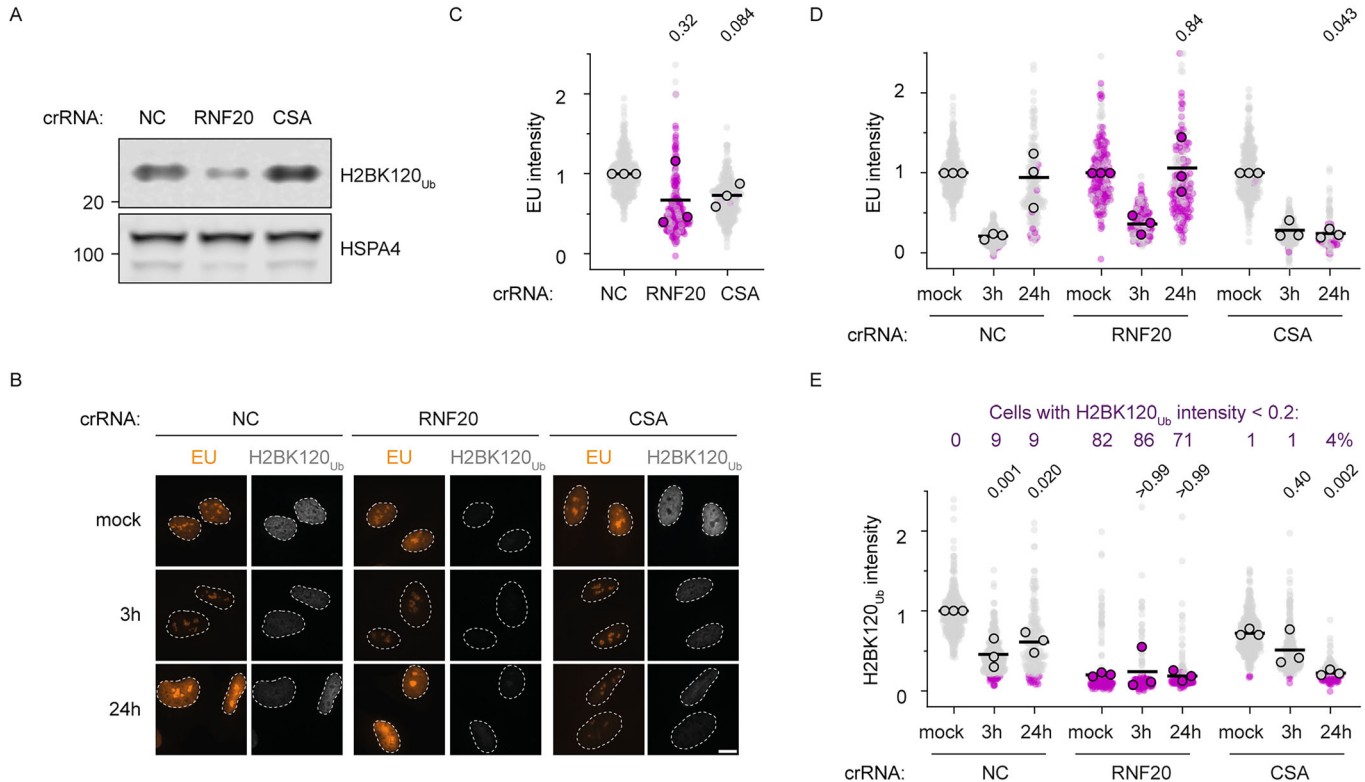

**Figure 3. H2BK120Ub is dispensable for transcription restart after DNA damage.**

(A) Western blot of RPE1 cells 5 days after Cas9 induction and transfection with indicated crRNAs. (B) Representative images of cells labeled for 1 h with EU, unirradiated or 3 h and 24 h after 12 J/m² UV-C, 5 days after transfection with indicated crRNAs, co-stained for H2BK120Ub. Dashed lines represent the nucleus defined by DAPI staining. Scale bar, 10 μm. (C) Nascent EU incorporation in nonirradiated cells transfected with indicated crRNAs relative to crNC transfected cells. Statistical significance on the means of three biological replicates was determined by a one-sample *t* test with hypothetical value 1. (D) Quantification of EU levels in conditions from (B), normalized to the mock condition per crRNA. Individual data points represent cells, circles represent means from biological replicates. Pink cells are cells with relative H2BK120Ub levels below 0.2 intensity. Statistical significance was determined by one-way ANOVA on the means of three biological replicates with Dunnett's multiple comparisons test against the respective mock condition. (E) Quantification of H2BK120Ub levels in conditions from (B), normalized to crNC mock condition. Pink cells are cells with relative H2BK120Ub intensity below 0.2. Numbers indicate the percentage of cells with relative H2BK120Ub intensity below 0.2. Statistical significance was determined by ordinary one-way ANOVA on the means of three biological replicates with Dunnett's multiple comparisons test against the respective mock condition. Source data are available online for this figure.

correlation. However, this does not necessarily demonstrate a causal relationship between this histone mark and transcription restart after UV irradiation. H2BK120Ub is deposited by the E2 conjugase RAD6 in conjunction with the heterodimeric E3 ligase RNF20-RNF40, which is essential for cell viability. In line with this, we were unable to generate viable RNF20KO clones.

To test whether H2BK120Ub contributes to transcription recovery after genotoxic stress, we performed conditional knockout of RNF20 using crRNF20 transfection combined with induction of Cas9 expression. This resulted in a reduction of H2BK120Ub within 5 days (Fig. 3A,B) without a significant effect on nascent transcription as measured by EU incorporation (Fig. 3C). More than 70% of RNF20-depleted cells had H2BK120Ub levels below 20% of that of control-transfected cells (purple dots in Fig. 3C–E). Despite this, RNF20-depleted cells recovered RNA synthesis normally after UV (Fig. 3D), indicating that with decreased H2BK120Ub levels transcription recovery can still happen to a similar extent as in WT cells. In agreement with previous reports (Mao et al, 2014; van den Heuvel et al, 2021), H2BK120Ub levels decreased 3 h after UV, coinciding with reduced transcription

(Fig. 3E). Upon recovery of transcription in control cells at 24 h post-UV, H2BK120Ub levels began to rise again, a recovery not observed in crCSA-treated cells (Fig. 3E). These findings suggest that H2BK120Ub and RNF20 are not required for transcription restart after UV-induced DNA damage. Instead, H2BK120Ub levels decrease as a consequence of transcription inhibition and are restored as transcription resumes.

## H2BK120 ubiquitylation levels correlate with transcription restart after UV

To explore this observation further, we measured transcription restart (via 5-EU labeling) and H2BK120Ub levels in WT and CSBKO RPE1 cells up to 48 h after UV irradiation. Following a pronounced decrease at 3 h after UV, WT cells exhibited full transcription recovery after 24 and 48 h, accompanied by a corresponding drop and subsequent restoration of H2BK120Ub levels. At 24 h, H2BK120Ub levels increased again and increased slightly further at 48 h (Fig. 4A–C). In CSBKO cells, transcription failed to recover at both 24 and 48 h, and H2BK120Ub levels remained as low as at 3 h

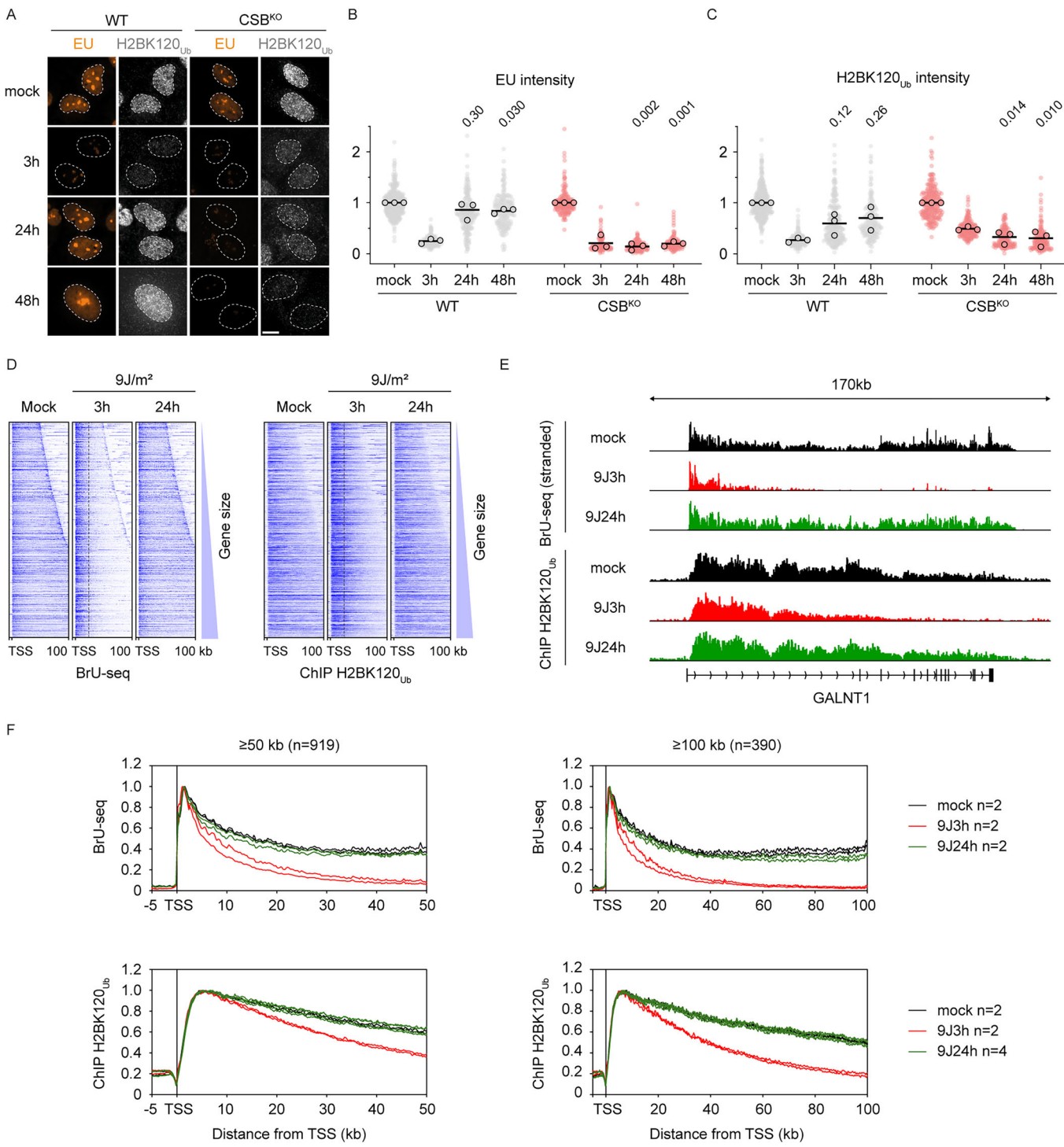

post-UV (Fig. 4A–C), suggesting that H2BK120Ub levels track transcription recovery, potentially with some delay.

To orthogonally validate these findings, we performed genome-wide ChIP-seq for H2BK120Ub and reanalyzed our previous BrU-seq dataset, which measures genome-wide nascent RNA synthesis (van der Weegen et al, 2021). BrU-seq in RPE1 cells revealed a strong decrease in nascent transcription within gene bodies 3 h after UV, consistent with photolesion frequency and the

likelihood of RNAPII encountering them (Perdiz et al, 2000). This effect is evident in heatmaps from transcription start sites (TSSs) to up to 100 kb for 919 genes of more than 50 kb sorted by gene length, a genome browser track for the GALNT1 gene, and a metagene analysis of 919 genes ≥50 kb or 390 genes ≥100 kb (Fig. 4D–F). H2BK120Ub followed a similar trend, although decreases were primarily detected beyond 20–30 kb, whereas nascent transcription was already sharply reduced beyond 10 kb

◀

**Figure 4.  H2BK120$_{Ub}$ levels correlate with transcription levels.**

(A) Representative images of cells labeled for 1 h with EU, unirradiated or 3, 24 h or 48 h after 12 J/m$^2$ UV-C, in WT and CSB$^{KO}$ cells, co-stained for H2BK120$_{Ub}$. Dashed lines represent the nucleus defined by DAPI staining. Scale bar, 10 µm. (B) Quantification of EU levels in conditions from (A), normalized to the mock condition per genotype. Individual data points represent individual cells, circles represent means from three biological replicates. Statistical significance was determined by a one-sample $t$ test with hypothetical value 1. (C) Quantification of H2BK120$_{Ub}$ levels in conditions from (A), normalized to the mock condition per genotype. Individual data points represent individual cells, circles represent means from three biological replicates. Statistical significance was determined by a one-sample $t$ test with hypothetical value 1. (D) Heatmaps of Bru-seq and H2BK120$_{Ub}$ ChIP-seq from the transcription start site (TSS) up to 100 kb of 919 genes of at least 50 kb in RPE1 WT cells after mock treatment, 3 h or 24 h after UV-C (9 J/m$^2$). Genes are ordered according to gene length. (E) Representative gene track showing read density of BrU and H2BK120$_{ub}$ signal across the GALNT1 gene in RPE1 WT cells after mock treatment, 3 h or 24 h after UV-C (9 J/m$^2$). (F) Metaplots of BrU and H2BK120$_{ub}$ signal in RPE1 WT cells after mock treatment, 3 h or 24 h after UV-C (9 J/m$^2$) of genes above 50 kb ($n = 919$, left panel) or above 100 kb ($n = 390$, right panel). For equal comparison of profiles, signals of each condition were normalized to their maximum readcount. Source data are available online for this figure.

(Fig. 4D–F). This suggests that H2B is not immediately deubiquitylated when nascent transcription is lost. Indeed, H2BK120$_{Ub}$ levels dropped to ~40% at 50 kb and ~20% at 100 kb into gene bodies, whereas nascent transcription had already fallen to background levels at these positions (Fig. 4D–F). At 24 h post-UV, both nascent transcription and H2BK120$_{Ub}$ levels fully recovered, even up to 100 kb. Thus, although H2BK120$_{Ub}$ is not required for transcription recovery after UV, the levels of this co-transcriptional mark correlate with transcription recovery, indicating that it is deposited behind RNAPII without being necessary for the restart itself.

## PAF1C subunits PAF1 and CTR9, but not RTF1, are required for transcription restart

Since the PAF1C has been shown to physically interact with RAD6/RNF20-RNF40 to stimulate H2BK120 ubiquitylation (Fetian et al, 2023; Kim et al, 2009; Van Oss et al, 2016), and H2BK120$_{Ub}$ in turn promotes H3K4$_{me3}$ and H3K79$_{me}$ deposition, we tested the levels of these marks after conditional knockout of PAF1C subunits CTR9, PAF1, and RTF1. Conditional knockout of CTR9, PAF1, and more prominently RTF1, resulted in a pronounced reduction in H2BK120$_{Ub}$ levels (Fig. 5A,B). In addition, H3K4$_{me3}$ levels were decreased, particularly in RTF1-depleted cells. Conditional knockout of CTR9, PAF1 and RTF1 also decreased the H2BK120$_{Ub}$-stimulated mark H3K79$_{me2}$ although less pronounced than the drop in H2BK120$_{Ub}$ (Fetian et al, 2023; Kim et al, 2009; Krogan et al, 2003; Valencia-Sanchez et al, 2019; Wood et al, 2003; Worden et al, 2019) (Fig. 5A,B), possibly due to the long half-life of H3K79$_{me}$ (Chory et al, 2019) or H2BK120$_{Ub}$-independent H3K79 methylation pathways. Together, these results confirm that these PAF1C subunits contribute to H2BK120$_{Ub}$, H3K4$_{me3}$, and H3K79$_{me}$ deposition.

Previous work revealed that the PAF1C subunit PAF1 is required for transcription restart after UV irradiation (van den Heuvel et al, 2021). Furthermore, RNAPII stably interacts with five PAF1C subunits (CTR9, PAF1, CDC73, LEO1, and SKI8), but not with RTF1, in response to UV-induced DNA damage (van den Heuvel et al, 2021). While PAF1 and RTF1 are part of the same complex in yeast, RTF1 does not stably associate with PAF1C in human cells (Francette and Arndt, 2024; Francette et al, 2021; Zumer et al, 2021). We therefore hypothesized that these subunits may have different contributions to transcription restart. In line with previous results using siRNAs or a degron system to deplete PAF1 in U2OS cells (van den Heuvel et al, 2021), we found that conditional knockout of PAF1 by crPAF1 transfection in RPE1 cells impaired transcription restart after UV irradiation (Fig. 5C,D).

Conditional knockout of another core subunit, CTR9, similarly impaired transcription restoration (Fig. 5C,D), suggesting this function is shared among core PAF1C subunits. In contrast, conditional knockout of the dissociable PAF1C subunit RTF1, confirmed by western blot, did not affect transcription recovery (Fig. 5C,D), suggesting that RTF1 is dispensable for this process. Together, these findings indicate that core PAF1C subunits, but not the dissociable subunit RTF1, drive transcription restoration after UV in a manner independent of H3K79$_{me}$, H3K4$_{me3}$, or H2BK120$_{Ub}$ deposition.

## The transient PAF1C-RNAPII interaction is stabilized by CSB after UV

Our previous pull-down of PAF1C from unirradiated cells, followed by mass spectrometry, revealed that PAF1C forms a stable complex comprising five subunits, without detectable interactions with RNAPII subunits (Fig. 6A). In contrast, pull-down of PAF1C from UV-irradiated cells using the same protocol followed by mass spectrometry revealed interactions with RNAPII subunits as well as CSA and CSB (Fig. 6B). These results suggest that the PAF1C-RNAPII interaction is transient during normal transcription, but becomes stabilized following UV irradiation (van den Heuvel et al, 2021). To investigate this further, we examined the relationship between PAF1C and RNAPII in more detail. Co-immunoprecipitation of elongating RNAPII from UV-irradiated RPE1 cells, which is suited for detecting stable interactors of RNAPII, showed that the interaction of PAF1C, but not RTF1, with RNAPII is enhanced by UV damage in a CSB-dependent manner (Fig. 6C,D), in line with observations in U2OS cells (van den Heuvel et al, 2021). No interactions of PAF1C or RTF1 with RNAPII were detected under undamaged conditions, likely because these interactions are too transient to be captured by co-IP (Francette et al, 2021).

Proximity labeling methods, such as those using the biotin ligase UltraID, allow detection of transient protein-protein interactions (Gonzalez-Vinceiro et al, 2025; Goos et al, 2022; Kubitz et al, 2022). To capture transient binding of elongation factors to RNAPII during unperturbed transcription, we endogenously fused UltraID to the elongation factor ELOF1, which interacts with the elongating RNAPII complex, via a knock-in approach (Geijer et al, 2021; van den Heuvel et al, 2021). As a control, UltraID was knocked into the ELOF1 locus, separated by a T2A ribosome-skipping sequence, producing separate ELOF1 and UltraID proteins from a single mRNA. Proximity-labeling using ELOF1-UltraID allowed the detection of several known transcription elongation factors,

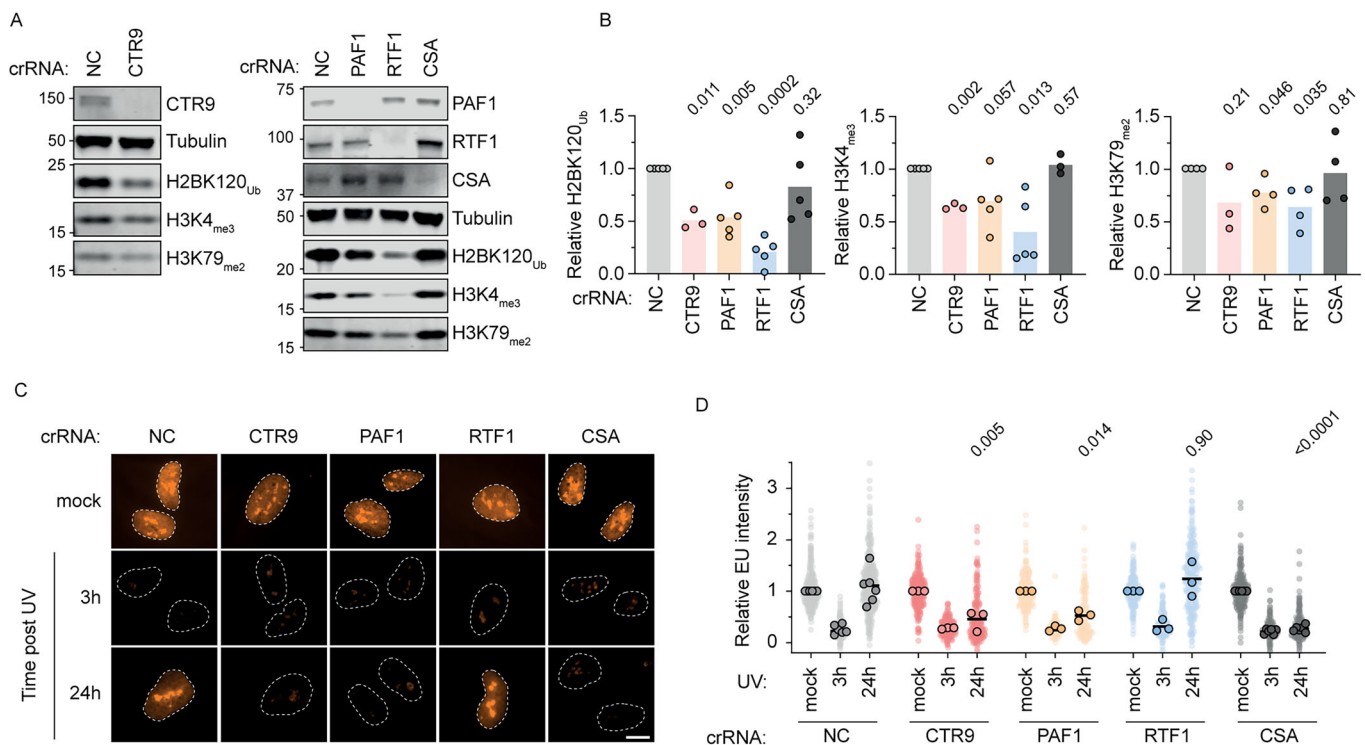

**Figure 5. RTF1 is not required for transcription restart.**

(A) Western blot of RPE1 cells 5 days after Cas9 induction and transfection with indicated crRNAs. (B) Quantifications of biological replicates of western blots of H2BK120_Ub (left), H3K4_me3 (middle), and H3K79_me2 (right) levels, normalized to loading control tubulin and relative to crNC (non-targeting control). Statistical significance was determined by a one-sample *t* test with hypothetical value 1. (C) Representative images of cells 5 days after transfection with indicated crRNAs labeled for 1 h with EU, either unirradiated or 3, or 24 h after 12 J/m$^2$ UV-C. Dashed lines represent the nucleus defined by DAPI staining. Scale bar, 10 μm. (D) Quantification of EU levels of conditions from (C). Cells are depicted as individual data points. Circles are individual means of three biological replicates. Lines indicate means. Statistical significance was determined by one-way ANOVA on the means of biological replicates against the crNC transfected 24 h after UV condition. Source data are available online for this figure.

including SPT5, SPT6, and IWS1 (Fig. EV2), validating this approach as a suitable method to detect transient interactions of the RNAPII elongation complex. Proximity-labeling revealed that PAF1C and RTF1 are in close proximity to the elongating RNAPII complex in the absence of DNA damage (Fig. 6E,F), while no interactions were detected in the UltraID-only control. Importantly, in CSB$^{KO}$ cells, we still detected the transient interaction of PAF1C and RTF1 with the elongating RNAPII complex, revealing that the PAF1C-RNAPII interaction during normal transcription is independent of CSB (Fig. 6F).

Together, these findings indicate that DNA damage induces a CSB-dependent stabilization of the PAF1C-RNAPII interaction, which is likely central to the transcription restart function of PAF1C. Overall, we demonstrate that the PAF1C elongation complex contributes to transcription restart after DNA damage independently of the RTF1 subunit and in a manner that does not depend on transcription-associated histone marks H3K79_me, H3K4_me3, or H2BK120_Ub.

## Discussion

Our study demonstrates that initial post-repair transcription restart is independent of the histone modifications H2BK120_Ub, H3K4_me3,

and H3K79_me. While these marks correlate with active transcription, they are not strictly required for transcription (Cao et al, 2020; Fetian et al, 2024; Ljungman et al, 2019; Wang et al, 2013; Wang and Helin, 2025). Our data indicate that RNA synthesis can occur in the absence of these marks, both under unchallenged conditions and following DNA repair.

Previous studies in model systems other than human cells have shown that transcription restart after DNA damage can partially depend on methylation of H3K4 or H3K79. In *C. elegans*, H3K4 methylation and the SET1/COMPASS complex facilitate transcription restart after DNA damage (Wang et al, 2020), whereas our findings show that neither H3K4_me3 nor the SET1/COMPASS complex is required for transcription recovery in human cells. Consistent with this, in *C. elegans*, only H3K4_me2, but not H3K4_me3, dynamically changes in response to DNA damage (Wang et al, 2020). H3K4_me2 and H3K4_me3 differ in dynamics, genomic distribution, and have at least partially non-overlapping functions (Pinskaya and Morillon, 2009; Wang et al, 2023; Wang and Helin, 2025). Furthermore, H3K4 methyltransferases other than SET1/COMPASS can contribute to H3K4_me2 deposition (Wang et al, 2020), potentially explaining interspecies differences. In addition, in the *C. elegans* experiments, L1 larvae were irradiated and followed during development to adulthood, suggesting that H3K4_me2 deposition may be a developmental response not

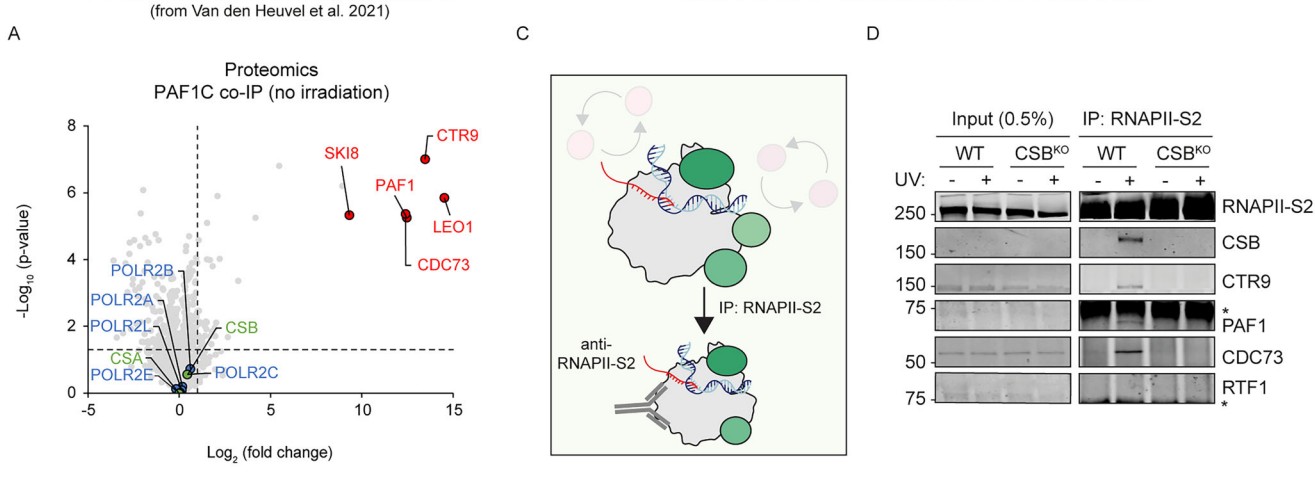

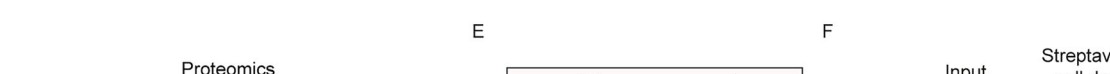

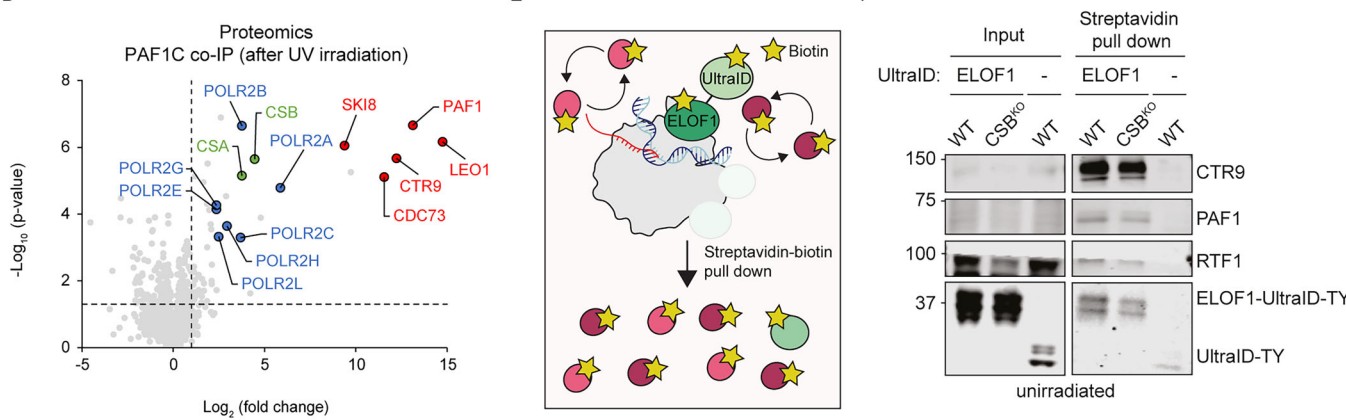

**Figure 6. The DNA damage-induced stable PAF1C-RNAPII interaction depends on CSB.**

(A) Mass spectrometry after co-immunoprecipitation (co-IP) of GFP-LEO1 in the absence of DNA damage (data adapted from (van den Heuvel et al, 2021)). Significance was determined by a two-sided *t* test. (B) Mass spectrometry after co-IP of GFP-LEO1 after UV-C irradiation (data adapted from (van den Heuvel et al, 2021). Significance was determined by a two-sided *t* test. (C) Cartoon depicting the RNAPII-S2 (ser2-phosphorylated CTD of RPB1) co-IP approach for the detection of stable interactors (depicted in green) of elongating RNAPII. (D) RNAPII-S2 co-IP in unirradiated (−) or 1 h after 12 J/m² UV-C irradiation in WT and CSB^KO cells. (E) Cartoon depicting the ELOF1-ultraID proximity labeling approach for the detection of transient interactors (depicted in red) of elongating RNAPII. (F) Streptavidin pull-down of ELOF1-UltraID proximity biotin-labeled proteins in WT and CSB^KO cells. ELOF1-T2A-UltraID (right lane; −) is a negative control. Source data are available online for this figure.

captured in cultured human cells (Wang et al, 2020). It is conceivable, for instance, that the initial round of immediate restart by RNAPII is independent of this histone mark, whereas sustained expression of previously damaged genes during development requires epigenetic marking by transcription-associated histone modifications.

In mouse embryonic fibroblasts, DOT1L-mediated H3K79 methylation has been reported to contribute to transcription restart (Oksenych et al, 2013), whereas we find no requirement for this modification in human cells. Similar to previously observed differences in TCR requirements between mouse and human cells (Apelt et al, 2020), this may reflect interspecies differences in chromatin prerequisites for transcription recovery. We also note that DOT1L-deficient mouse cells were generated using a gene-trap insertion, and the transcription restoration phenotype was not

rescued by re-expression of DOT1L (Oksenych et al, 2013; Steger et al, 2008), leaving open the possibility of off-target effects. A previous report in human cells identified a role for DOT1L in transcriptional restart at transcription-replication conflict sites (Werner et al, 2025), suggesting that dependency on DOT1L may be context- and location-specific.

Post-repair transcription restart can occur without histone mark deposition. Core PAF1C subunits PAF1 and CTR9 are essential for efficient transcription restart, whereas the dissociable subunit RTF1 is dispensable. This aligns with observations that RTF1 dissociates from RNAPII before CSB binding in vitro (Kokic et al, 2021), while the other five subunits are stabilized on RNAPII after DNA damage in vivo (van den Heuvel et al, 2021). Because RTF1 is essential for PAF1C-dependent histone modifications, this suggests that PAF1C's role in transcription recovery is independent of these

marks. Interestingly, RTF1 contributes to RNAPII velocity (Zumer et al, 2021), indicating that not all transcription elongation factors are required for transcription restart. PAF1 may influence restart via regulation of RTF1-independent transcription elongation (Hou et al, 2019; Rondon et al, 2004; Zumer et al, 2021) or promoter-proximal pause release (Chen et al, 2015b; Chen et al, 2017; Yu et al, 2015). Furthermore, PAF1C physically interacts with CSB (Tiwari et al, 2021; van den Heuvel et al, 2021), and PAF1C's interaction with RNAPII depends on CSB, suggesting a dedicated TCR-specific pathway for post-repair transcription activation.

In conclusion, our study provides evidence that transcription recovery following genotoxic stress depends on PAF1C but is independent of transcription-associated histone modifications and RTF1. It will be interesting to further dissect the specific requirements of post-repair transcription restart compared to general transcription.

# Methods

## Reagents and tools table

| Reagent/resource | Reference or source | Identifier or catalog number |
|---|---|---|
| **Experimental models** | | |
| 48BR-hTERT | Apelt et al, 2021 | NA |
| RPE1-iCas9 (WT) | van der Weegen et al, 2021 | NA |
| RPE1-iCas9 CSB^KO (1–15) | van der Weegen et al, 2021 | NA |
| RPE1-iCas9 DOT1L^KO2 | This study | NA |
| RPE1-iCas9 DOT1L^KO6 | This study | NA |
| RPE1-iCas9 KI ELOF1 -ultraID-TY #3 | This study | NA |
| RPE1-iCas9 KI ELOF1 T2A-ultraID-TY #4 | This study | NA |
| RPE1-iCas9 CSB^KO (1–15) KI ELOF1-UltraID #2-10 | This study | NA |
| **crRNAs** | | |
| OR10A7 (NC) TTTCTGTGGACCAAATGCG | IDT | crML#001_crOR10A7_TKOv3#4 |
| DOT1L-6 (KO6) GATATGCGCGCAGGAGTCCAG | IDT | crR#120_DOT1L_TKOv3#2 |
| RBBP5 TTTCAGGCGACTG TGACCAG | IDT | crML#006_ crRBBP5_TKOv3#4 |
| RNF20-4 (combined with RNF20-8 to improve KO efficiency) GTGGAA ACAATTAAGCTAGG | IDT | crML#004_crRNF20_TKOv3#1 |
| RNF20-8 (combined with RNF20-4 to improve KO efficiency) TTTCAGGCGA CTGTGACCAG | IDT | crML#008_crRNF20_TKOv3#3 |
| CSA GGAGAGCAGAGTCAACACGG | IDT | crML#007_crCSA_TKOv3#3 |
| RTF1 GCTCAGCTCACCTTCCTCAG | IDT | crML015_RTF1_TKOv3-2 |
| PAF1 TGTGAAGCAGCAGTTTACCG | IDT | crML#009_PAF1_TKOv3_#4 |
| CTR9-11 (combined with CTR9-14 to improve KO efficiency) TAGCTGGAATACTACAAGCA | IDT | crML#011_CTR9_TKOv3_#2 |

| Reagent/resource | Reference or source | Identifier or catalog number |
|---|---|---|
| CTR9_14 (combined with CTR9-11 to improve KO efficiency) AAAGCATTGCGTACTAACCC | IDT | crML#014_CTR9_TKOv3#1 |
| **Plasmids** | | |
| pX458-(Cas9-2A-GFP)_ sgDOT1L_KO-2 | This study, based on Addgene plasmid #48138 | pML#380 |
| H.Arm1_ELOF1_T2A _ultaID_H.Arm2 | This study, based on Addgene plasmid #72824 | pML#355 |
| H.Arm1_T2A_ELOF-_T2A_ultaID_H.Arm2 | This study, based on Addgene plasmid #72824 | pML#385 |
| pX458-(Cas9-2A-GFP)-sgELOF1-3 | This study, based on Addgene plasmid #48138 | pML#187 |
| **Oligonucleotides** | | |
| CACCGTGATTGGATAGAC GCCTGCG | IDT | OML#302_sgELOF1-3a - sgRNA-ELOF1-Sense for cloning pml#187 |
| AAACCGCAGGCGT CTATCCAATCAC | IDT | OML#303_sgELOF1-3b - sgRNA-ELOF1-Antisense for cloning pml#187 |
| GCTGCGTGGTTTCCTCAAAC | IDT | OML#1217_ELOF1_seq_fw3 - conformation plasmid |
| ATGTTGCCCAGGCTGGTATC | IDT | OML#320_sgELOF1-2_Seq_FW gDNA PCR |
| CGGGAAGTCCAGTTGAGATG | IDT | OML#323_sgELOF1-3_Seq_RV gDNA PCR |
| CACCGTGATTGGA TAGACGCCTGCG | IDT | OML#302 - nested PCR gDNA ELOF1 |
| GCCACCGGTCCGCGG TTTAAACTTAAGCT | IDT | OML#674 - nested PCR gDNA ELOF1 |
| GCACAATTGGC ATCAAGTAATC CTCTCACCTTAGCTTCCC | IDT | OML#693_homology_arm1_ELOF1 |
| GACGTCGACTCTGT GTCGTCTCTG ATTGGCCGTCTCGCAGGC | IDT | OML#694_homology_arm1_ELOF1 |
| GCACGTACGACCCGCC CCCTGAGCA GCCCCGCGTACTGTGG | IDT | OML#695_homology_arm2_ELOF1 |
| GACCATATGGAAGGCC AGAGGTCAGC GTCTGGCTAGAGG | IDT | OML#696_homology_arm2_ELOF1 |
| CAGGAAACAGCTATGAC | IDT | OML#172_AID-N/B_flank2 Rv |
| **Chemicals, enzymes, and other reagents** | | |
| 5-ethynyl-uridine (EU) | VWR | CLK-N002-10 |
| Doxycline | Clontech | 8634-1 |
| Lipofectamine RNAimax | Invitrogen | 13778150 |
| DOT1Li (Pinometostat, EPZ5676) *final concentration used 5 μM* | Selleckchem | S7062 |
| RNAPi (BMH-21) *final concentration used 1 μM* | Gift from Julian Stingele | NA |
| RNAPIIi (DRB) *final concentration used 100 μM* | Sigma | D1916-50MG |
| RNAPIIIi (ML-60218) *final concentration used 100 μM* | Gift from Julian Stingele | NA |
| **Antibodies** | | |
| Rabbit anti-CSA/ERCC8 | Abcam, #137033 (EPR9237) | aML#028 |
| Mouse anti-α-Tubulin | Sigma, #T6199 (DM1A) | aML#008 |

| Reagent/resource | Reference or source | Identifier or catalog number |
|---|---|---|
| Rabbit anti-HSPA4 | Novus Biologicals, NBP1-81696 | aML#114 |
| Rabbit anti-H3K4me$_3$ | Cell Signaling, #9751S | aML#237 |
| Rabbit anti-DOT1L | Cell Signaling, #77087 (D1W4Z) | aML#216 |
| Rabbit anti-H3K79me$_2$ | Abcam, ab3594 | aML#227 |
| Rabbit anti-H2B-K120$_{Ub}$ | Cell Signaling, 5546S | aML#177 |
| Rabbit anti-PAF1 | Bethyl, A300-172A | aML#022 |
| Rabbit anti-RTF1 | Bethyl, A300-178A | aML#257 |
| Rabbit anti-RNAPII | Bethyl, A304-405A | aML#088 |
| Rabbit anti-RNAPII-S2 | Abcam, ab5095 | aML#024 |
| Mouse anti-SPT5 | Santa Cruz, sc-133217 | aML#122 |
| Rabbit anti-CTR9 | Bethyl, A301-395A | aML#049 |
| Rabbit anti-IWS1 | Cell Signalling, #5681 | aML#270 |
| Rabbit anti-SPT6 | Protein tech, 23073-1-AP | aML#236 |
| Rabbit anti-PPP1CB | Abcamab53315 | aML#287 |
| Mouse anti-TY1 | Diagenode, C15200054 | aML#154 |
| Goat anti-Rabbit Alexa 647 | Thermo Fisher, A-21245 | aML#016 |
| CF770 Goat Anti-Mouse IgG (H + L) | Biotium, VWR #20077 | aML#009 |
| **Antibodies** | | |
| Graphpad Prism 10.4.1 | NA | NA |
| ImageJ v.1.48 | NA | NA |

## Cell lines and cultures

The RPE1 and 48BR cell lines were cultured at 37 °C in an atmosphere of 5% $CO_2$ in DMEM GlutaMAX (Thermo Fisher Scientific) supplemented with penicillin/streptomycin (Sigma), and 8–10% fetal bovine serum (FBS; Bodinco BV or Thermo Fischer Scientific (Gibco)).

## Generation of knockout cells

Parental RPE1-hTERT cells stably expressing inducible Cas9 (iCas9) that are also knockout for TP53 and the puromycin-N-acetyltransferase PAC1 gene were described previously (referred to as RPE1-iCas9) (van der Weegen et al, 2021). To generate knockouts in RPE1-iCas9, one day after induction of Cas9 by addition of 200 ng/ml doxycline, cells were transfected with 10 nM crRNA:tracrRNA duplexes using 1:1000 Lipofectamine RNAimax transfection reagent. Cell lines and crRNAs are listed in the Reagents and Tools Table. Two days after transfection, single cells were plated by limiting dilution. After single-cell clone expansion, clones were selected by western blot and Sanger sequencing.

## Conditional knock-out assays

To conditionally deplete essential genes, we transfected RPE1-iCas9 cells with 10 nM crRNA:tracrRNA duplexes with 1:1000 Lipofectamine RNAimax, one day after induction of Cas9 by the addition of 200 ng/ml doxycycline. crRNAs are listed in the Reagents and Tools Table. We performed experiments 5 days after transfection on the pools of cells which contain a mixed population of mutations in the target gene. Effective depletion was assessed 5 days after transfection by western blot and/or immunofluorescence.

## Western blotting

Total cell lysates were harvested by scraping cells in Laemmli-SDS sample buffer. Chromatin fractions were obtained by lysing cell pellets on ice in EBC-1 buffer (50 mM Tris [pH 7.5], 150 mM NaCl, 2 mM MgCl2, 0.5% NP-40, and protease inhibitor cocktail (Roche)) for 20 min at 4 °C, on a rotating wheel, followed by centrifugation and removal of the supernatant. Chromatin pellets were resuspended in Laemmli-SDS sample buffer. Total cell lysates or chromatin fraction samples were boiled for 10 min at 95°C. Proteins were separated on Criterion™ XT Tris-Acetate 3-8% Protein Gels (BioRad, #3450131) in Tris/Tricine/SDS Running Buffer (BioRad, #1610744) or on Criterion Xt bis-tris 4–12% gels in MOPS running buffer. Then, blotted onto PVDF membranes (IPFL00010, EMD Millipore) in Tris/glycine blotting buffer (0.025 M Tris, 0.192 M glycine) with 20% methanol. Membranes were blocked with 5% fat-free milk in PBS with 0.1% Tween-20 for 1 h at room temperature. Membranes were then probed with indicated antibodies in 5% fat-free milk in PBS with 0.1% Tween-20 (Antibodies are listed in the Reagents and Tools Table). Proteins were stained with fluorochrome-conjugated secondary antibodies and were detected on an Odyssey CLx system and Image Studio software (Li-Cor).

## Recovery of RNA synthesis (RRS)

After irradiation with UV-C light (12 J/m$^2$), cells were allowed to recover for indicated times followed by pulse labeling with 400 μM 5-ethynyl-uridine (EU) for 1 h. Where indicated, cells were treated with inhibitors listed in the Reagents and Tools Table. Next, cells were medium chased with DMEM without supplements, followed by fixation in 3.7% paraformaldehyde in PBS for 15 min. Cells were permeabilized with 0.5% Triton X-100 for 10 min and blocked in 1.5% bovine serum albumin (BSA, Thermo Fisher) in PBS. Nascent EU-labeled RNA was visualized by click-it chemistry, by incubating the cells with 60 μM Atto azide-Alexa 594 (Atto Tec), 4 mM copper sulfate (Sigma), 10 mM ascorbic acid (Sigma), and 0.1 μg/mL DAPI in 50 mM Tris-buffer pH 8 for 1 h. After washing extensively with PBS, slides were mounted in Polymount (Brunschwig).

## Microscopic analysis of fixed cells

Images of fixed samples were acquired on a Zeiss AxioImager M2 widefield fluorescence microscope equipped with ×63 PLAN APO (1.4 NA) oil-immersion objectives (Zeiss) and an HXP 120 metal-halide lamp used for excitation. Fluorescent probes were detected

using the following filters for DAPI (excitation filter: 350/50 nm, dichroic mirror: 400 nm, emission filter: 460/50 nm), Alexa 555/594 (excitation filter: 545/25 nm, dichroic mirror: 565 nm, emission filter: 605/70 nm), or Alexa 647 (excitation filter: 640/30 nm, dichroic mirror: 660 nm, emission filter: 690/50 nm). Images were recorded using ZEN 2012 software (blue edition, version 1.1.0.0) and analyzed in ImageJ (1.48 v).

## Generation of ELOF1-UltraID cells

To generate endogenous knock-ins of UltraID-TY1 or T2A-UltraID-TY1 into the ELOF1 locus, RPE1-iCas9 WT and CSB[KO] (1–15) cells were cultured in 6 cm plates prior to transfection. To promote homology-directed repair, cells were pretreated with PolQ inhibitor (ART558, MCE, 10 μM) and DNA-PK inhibitor (NU7441, MCE, 2 μM) 24 h before transfection. RPE1 WT or CSB[KO] (1–15) cells were then transfected with plasmids encoding a single guide RNA (sgRNA) targeting ELOF1, a Cas9-GFP construct (pML#187), and knock-in plasmids containing 1 kb homology arms targeting the ELOF1 locus, either ELOF1-UltraID-TY1-T2A-Puromycin (pML#355) or T2A-UltraID-TY1-T2A-Puromycin (pML#385), using 1 μg/μl PEI (Merck) in OptiMEM (Life Technologies, 51985034), followed by overnight incubation. Both plasmids (pML#355 and pML#385) contain a puromycin resistance cassette flanked by ~1 kb homology arms to enable selection (Plasmids are listed in Reagents and Tools Table). After overnight transfection, cells were cultured for 2–3 days in standard DMEM supplemented with PolQ (10 μM) and DNA-PK (2 μM) inhibitors. GFP-positive cells were subsequently sorted by FACS and seeded. Single clones were isolated, transferred to individual culture plates, and selected for ELOF1-UltraID knock-ins using 1 μg/ml puromycin. Genomic DNA was extracted, and successful knock-in of UltraID or T2A-UltraID at the ELOF1 locus was confirmed by Sanger sequencing following nested PCR amplification of genomic DNA.

## Proximity labeling and streptavidin-based enrichment of biotinylated proteins

Cells were seeded and grown in 15-cm dishes until reaching 80% confluency. They were then incubated with 50 μM biotin for 15 min, followed by a 15-min starvation in plain DMEM medium (Gibco, 31966-047). Subsequently, cells were collected by trypsinization and resuspended in 1× PBS at 4 °C. Immediately after resuspension, cells were pelleted by centrifugation at 1500 rpm and washed with 1× PBS. Cell pellets were snap-frozen in liquid nitrogen and stored at −80 °C until pull-down. Cells were lysed at 4 °C for 20 min using 1 ml of lysis buffer (50 mM Tris [pH 7.5], 150 mM NaCl, 0.1% deoxycholate, 2 mM MgCl$_2$, 0.5% NP-40, and protease inhibitor cocktail [Roche]) supplemented with 500 U/ml benzonase (Novagen). After lysis, the volume was adjusted to 2.5 ml with lysis buffer. To remove free biotin and cell debris, the protein lysate was processed using a G25/PD-10 size exclusion column (Cytiva, 17085101) according to the manufacturer's protocol. The resulting 3.5 ml protein eluate was incubated overnight at 4 °C with 30 μl streptavidin-sepharose beads (Merck Millipore, 69203-3) and 1000 U/ml benzonase to capture all biotin-labeled proteins. After six washes with washing buffer (50 mM Tris [pH 7.5], 300 mM NaCl, 0.1% deoxycholate, 1 mM EDTA, 0.5% NP-40), samples were boiled in 2× Laemmli buffer containing 0.2 mg/ml biotin and analyzed by western blot.

## RNAPII-S2 co-immunoprecipitation

Where indicated, cells were irradiated with UV-C (12 J/m²) and harvested 1 h after irradiation. Cells were lysed for 20 min at 4 °C in 1 mL EBC-1 buffer (50 mM Tris [pH 7.5], 150 mM NaCl, 2 mM MgCl$_2$, 0.5% NP-40, and protease inhibitor cocktail (Roche)), followed by collection of the chromatin-enriched pellet by centrifugation. Pellets were resuspended in EBC-1 buffer with 500 U/mL Benzonase Nuclease (Novagen) and 2 μg RNAPII-S2 1 h at 4 °C. Next, the salt concentration was increased to 300 mM NaCl. Samples were incubated for an additional 30 min at 4 °C, followed by centrifugation at 14,000 × g. In total, 50 μL of supernatant was mixed with 50 μL 2x Laemmli buffer as input sample. The remaining supernatant was incubated with 20 μL equilibrated Protein A agarose beads (Millipore) at 4 °C for 90 min (except for GFP-trap agarose incubated samples). Next, beads were washed six times with EBC-2 buffer (50 mM Tris, pH 7.5, 300 mM NaCl, 0.5% NP-40, 1 mM EDTA), and boiled in Laemmli-SDS buffer, followed by analysis by western blot.

## ChIP-seq

Cells were grown to 80–90% confluency and crosslinked with 0.5 mg/mL disuccinimidyl glutarate (DSG; Thermo Fisher) in PBS for 45 min at room temperature (RT). Cells were washed once with PBS followed by incubation with 1% formaldehyde for 20 min at RT. Fixation was stopped by the addition of glycine in PBS to a final concentration of 0.1 M for 3 min at RT. This was followed by washing with cold PBS and collection of the cells in 0.25% Triton X-100, 10 mM EDTA (pH 8.0), 0.5 mM EGTA (pH 8.0), and 20 mM Hepes (pH 7.6) in miliQ. Chromatin was pelleted by centrifugation for 5 min at 400 × g and incubated in 150 mM NaCl, 1 mM EDTA (pH 8.0), 0.5 mM EGTA (pH 8.0) and 50 mM Hepes (pH 7.6) in miliQ for 10 min at 4 °C. Chromatin was again pelleted by centrifugation and resuspended in ChIP-buffer (0.15% SDS, 1% Triton X-100, 150 mM NaCl, 1 mM EDTA (pH 8.0), 0.5 mM EGTA (pH 8.0), and 20 mM Hepes (pH 7.6) in miliQ) to a final concentration of 15 × 10⁶ cells/mL. Chromatin was sonicated to approximately one nucleosome using the Bioruptor Pico (Diagenode), 10 cycles of 30 s ON/30 s OFF in a 4 °C water bath. ubH2B ChIP was performed using ~28 μg of chromatin with 3 μl of ubH2B antibody (Cell Signaling; #5546S) by overnight incubation at 4 °C. Protein-chromatin-pull down followed with a 1:1 mix of protein A and protein G Dynabeads (Thermo Fisher; 10001D, 10003D). ChIP samples were extensively washed and purified using the Qiagen MinElute kit. Sample libraries were prepared using Hifi Kapa sample prep kit and A-T mediated ligation of Nextflex adapters or xGen UDI-UMI adapters. Samples were sequenced using an Illumina NovaSeq X, using paired-end sequencing with 150 bp from each end.

## BrU-seq

We re-analyzed our previously generated BrU-seq data already deposited under GSE149760 (samples GSM4511916, GSM4511917, GSM4511919, GSM4511916, GSM4511917, GSM4511919) (van der Weegen et al, 2021).

## ChIP-seq and BrU-seq data analysis

Sequences were trimmed using TrimGalore (Version 0.6.5). For ChIP-seq, reads were aligned to the GRCh38.p14 primary

assembly using bwa-mem tools (BWA,Version 0.7.17) (Li, 2013). Only uniquely mapping and high-quality reads (> q30) were included in the analyses. Duplicate reads were removed using Samtools (Version 1.11) with fixmate -m and markdup -r settings. For BrU-seq data, reads were aligned to the same genome using STAR (v2.7.7a) (Dobin et al, 2013) with settings --outFilterMulti-mapNmax 20 and --outSAMmultNmax 1 to allow up to 20 multimaps, but keeping only the primary map. Bam files were converted into TagDirectories (with fixed fragment length 150) (TagDirectories were stranded for BrU-seq) and UCSC genome tracks using HOMER tools (Version 4.8.2) (Heinz et al, 2010). Example genome tracks were generated in IGV (Version 2.4.3). A list of 49,948 gene-coordinates of GRCh38 was obtained from the UCSC genome database, selecting the "knownCanonical" table containing the canonical transcription start sites per gene (Karolchik et al, 2004). To prevent contamination of binding profiles, genes from this list were selected to be non-overlapping with at least 2 kb between genes and a minimal size of 3 kb ($n = 9944$). The selected gene coordinates were updated to GRCh38.p14. From this, a set of 3000 actively transcribed genes was selected by calculating read-densities of BrU-seq data in the first 3 kb of the gene in WT cells, using the AnnotatePeaks.pl tool of HOMER with default settings. From these 3000 actively transcribed genes, subselections of genes were defined based on gene size, as indicated per analysis. Read-density profiles around TSS coordinates were defined using the AnnotatePeaks.pl tool of HOMER (using stranded counts for BrU-seq), followed by normalizing the averaged metaprofiles to their internal maximum intensity (around the TSS). Individual datasets were subsequently processed into heatmaps or binding profiles using R (Version 4.3.1) and RStudio (Version 1.1.423).

## Data availability

H2BK120$_{Ub}$ ChIP-seq data is deposited on GEO with accession number: GSE311787.

The source data of this paper are collected in the following database record: biostudies:S-SCDT-10_1038-S44319-026-00761-0.

## Peer review information

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

## Acknowledgements

MSL laboratory was supported by the European Research Council Consolidator Grant STOP-FIX-GO (grant agreement No 101043815) and the Netherlands Scientific Organization (NWO) Vici grant (VI.C.212.005). The funders had no role in study design, data collection and analysis, the decision to publish, or the preparation of the manuscript.

## Author contributions

**Janne J M van Schie**: Conceptualization; Resources; Data curation; Formal analysis; Validation; Investigation; Visualization; Methodology; Writing—original draft; Project administration; Writing—review and editing. **Bram A F J de Groot**: Conceptualization; Data curation; Formal analysis; Validation; Investigation; Visualization; Methodology; Writing—original draft. **Diana van den Heuvel**: Conceptualization; Data curation; Formal analysis; Investigation; Visualization; Methodology. **Martijn S Luijsterburg**: Conceptualization; Supervision; Funding acquisition; Visualization; Writing—original draft; Project administration; Writing—review and editing.

Source data underlying figure panels in this paper may have individual authorship assigned. Where available, figure panel/source data authorship is listed in the following database record: biostudies:S-SCDT-10_1038-S44319-026-00761-0.

## Disclosure and competing interests statement

The authors declare no competing interests.

# Expanded View Figures

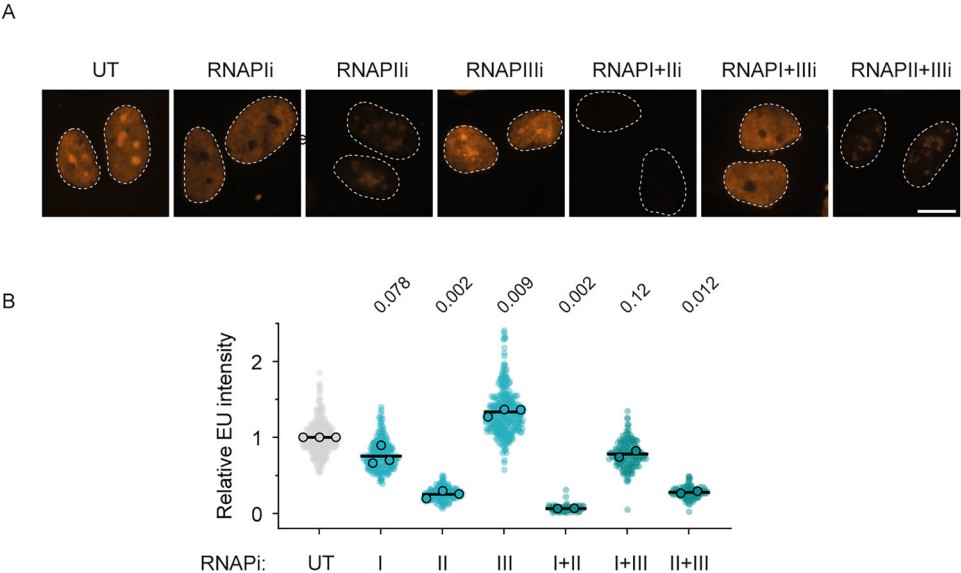

Figure EV1. **Contribution of RNAPI, RNAPII, and RNAPIII to nascent transcription.**

(**A**) Representative images of RPE1 cells labeled for 1 h with EU after treatment for 4 h with RNAPIi (BMH-21, 1 µM), RNAPIIi (DRB, 100 µM), and RNAPIIIi (ML-60218, 20 µM). Dashed lines represent the nucleus defined by DAPI staining. Scale bar, 10 µm. (**B**) Quantification of EU levels in conditions from (**A**). Statistical significance was determined by one-way ANOVA on the means of three biological replicates. Source data are available online for this figure.

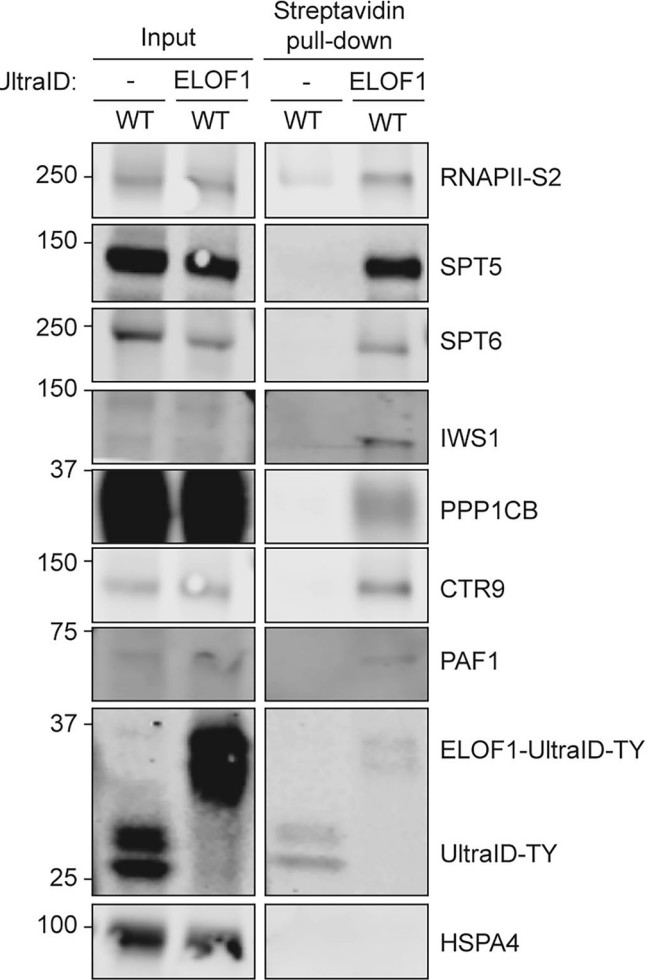

**Figure EV2.  ELOF1-ultraID proximity labeling detects transiently interacting transcription elongation factors.**

ELOF1-UltraID proximity biotin-labeled proteins were streptavidin precipitated followed by detection with western blot. ELOF1-T2A-UltraID (left lane; -) is a negative control. HSPA4 antibody is used as a non-elongation factor control. Source data are available online for this figure.

