## [Peer Review File · EMBO Reports]

PAF1C restores transcription after DNA damage independently of promoting histone mark deposition

Janne van Schie, Bram de Groot, Diana van den Heuvel, and Martijn Luijsterburg

Corresponding author(s): Martijn Luijsterburg (S.M.Luijsterburg@lumc.nl)

Review Timeline:

Submission Date:	16th Aug 25
Editorial Decision:	21st Aug 25
Revision Received:	5th Jan 26
Editorial Decision:	11th Feb 26
Revision Received:	3rd Mar 26
Accepted:	23rd Mar 26

Editor: Esther Schnapp

Transaction Report: The first round of review of this manuscript was performed at another journal.

Referee: 1

In this manuscript, van Schie et al. follow up on prior work from the Luijsterburg group (van den Heuvel et al., Nat. Commun. 2021), which showed a requirement for PAF1 in restarting transcription after DNA damage in U2OS cells. In the current manuscript, the authors investigate if deposition of PAF1-dependent histone marks is required for transcription restart post-UV irradiation in RPE1-hTERT cells. Using CRISPR-based strategies to induce mutations or knock out genes for DOT1L, RBBP5, and RNF20, the authors generate cell lines to test the roles in transcription restart for H3K79Me2, H3K4Me3, and H2BK120Ub, respectively. They also generate cell lines to deplete members of the PAF complex: PAF1, RTF1, and CTR9. Transcription restart after UV irradiation is assayed by EU-labeling of cells, and histone modifications are measured in the same cells by immunofluorescence. A final set of experiments tests association of PAF subunits with Pol II and elongation factor ELOF1.

The primary new conclusion from this manuscript is that transcription restart after DNA damage appears to occur normally in the absence of PAF-dependent histone modifications, and therefore, some other function of PAF1, which remains undefined, is required to restore transcription elongation after repair of DNA damage. In general, the data are convincing, although limited in depth, and they support the argument that no single PAF-dependent histone mark can explain PAF1's role in transcription restart. The authors should address the following comments and also a discrepancy between their current data and published work.

Specific comments:

1. Statistics are lacking for the graphs.

These can be added.

2. Figure 1. Unlike the other histone marks, H3K79Me2 levels in the individual cells are not shown. This information should be provided. Also, why do the authors focus on the dimethylation state?

Unlike the other marks, DOT1L can be knocked out and H3K79me is fully lost. It is therefore not necessary and adds nothing to measure H3K79me levels in individual cells. We show by western blot that the mark is lost. Also, there is not commercial antibody for H3K79me that works for immunostaining. The group of Fred van Leeuwen has developed an antibody, which requires harsh denaturation, which is incompatible with EU labeling and detection.

3. Figure 1d. Nascent transcription appears to be decreased in nonirradiated DOT1L-KO cells. A comment is needed.

This difference is not significant. We will add this information and comment if needed.

4. Figure 2A. A blot showing depletion of RBBP5 is needed. In addition, the levels of H3K4Me2 should be tested, especially since previous work has shown different effects of H3K4Me2 and H3K4Me3 on DNA repair.

We disagree. We show loss of H3K4me3 by western blot and by immunostaining. What does it add to buy and expensive RBBP5 antibody and show loss of this protein, when we already show the histone mark is gone? The transcription-associated mark linked to PAF1 function is H3K4me3, not me2. Moreover, both are fully dependent on RBBP5. We could, in principle,

buy an H3K4me2 antibody and show this mark is lost as well, but this adds very little and this has already been documented.

5. Figure 2D. The authors state that H3K4Me3 intensity is not changed in the CSA crRNA cells. This conclusion does not reflect the data at the 24-hour timepoint, where H3K4Me3 levels seem to be lower than for the NC-treated cells. Applying the $< 0.2\%$ cutoff seems quite stringent.

We can check if H3K4me3 levels are different at 24 h, but this is likely not significant. If it is, we will comment on it. The cut-off is 0.2 (%), meaning less than 20% of the intensity in crNC cells. We see how this annotation might be confusing and we will change it.

6. Figure 3A. Depletion of RNF20 should be confirmed by western blot.

What would this add? We measure H2B-Ub by western blotting and immunostaining at the single cell levels, where we correlate this to nascent transcript levels (measured by EU labelling)?

7. Figure 3D. In the van den Heuvel et al. paper, H2BK120Ub was shown to recover on gene bodies 8 hours post-UV treatment. In Figure 3D, H2BK120Ub levels are not recovered after 24 hours in the NC crRNA-treated cells. Because H2BK120Ub is a mark of active transcription, it is surprising to see recovery of transcription without recovery of this mark. Their previous observations seem more likely.

We have not shown that H2B-Ub levels recover in our previous paper (<https://www.nature.com/articles/s41467-021-21520-w>). See below: H2B-Ub levels lag behind non-irradiated cells at 8 h in WT cells, and even more strongly so in CSB-KO cells. This was done in U2OS cells.

Our current findings in RPE1 cells match these findings well. We detected reduced H2B-Ub levels at 3h and 24h, and even lower levels in CSA-KO at 24 h after UV. This results in similar to our previous findings.

We now take this further and instead of correlate, we address a causative relation between H2B-Ub and transcription recovery. We find that cells with less than 20% H2B-Ub level still recover transcription similar to WT cells.

The authors should provide an orthologous measure of H2BK120Ub in their crRNA-treated cells, such as CHIP-seq (as in van den Heuvel et al.) and also explain the discrepancies in their results.

There is no discrepancy. Our previous and current results align well. We see little added value in an orthologous measure of H2B-Ub levels by CHIP-seq since this method is not quantitative. Moreover, since CHIP is a bulk method, we cannot correlate histone mark levels and nascent transcript levels in single cells, as we do here by immunostaining. However, we could perform bulk CHIP-seq on H2B-Ub if so desired, with the objections we note above.

8. The authors have switched cell lines from their earlier study. An explanation for choice of cell line is needed.

We previously used U2OS cells, which are aneuploid cancer cells with high transcription levels and highly variable transcription recovery. We switched to RPE1 cells, which are near-diploid with transcription levels similar to primary fibroblasts and much more reproducible transcription recovery between experiments.

9. Figure 4E. This is essentially a repeat of Figure 2 in the van den Heuvel et al. paper. The only significant difference is showing RTF1 is not associated with Pol II after UV treatment. In our previous paper we used U2OS cells. This experiment is done in RPE1 cells. It is not an exact repeat, but the result between U2OS and RPE1 cells is indeed identical: CSB is essential to recruit the PAF1 complex subunits PAF1 and CTR9. We now show that RTF1 is not recruited after UV irradiation.

10. Figure 4F. The goal of this experiment is not well-described, but I believe the authors are primarily testing if PAF subunits CTR9 and PAF1 can be detected in the elongation complex under normal conditions, which is something that has been well documented in many experiments.

Our description is: *“Our co-immunoprecipitation approach fails to detect PAF1 or RTF1 interactions with RNAPII in undamaged conditions, while these are well described interactions³⁴. As mentioned, this is likely because the interactions are too transient and therefore not detected using our IP approach. Proximity labelling methods, such as those using biotin ligase UltraID, allow detection of transient protein-protein interactions⁵⁵. **To study transient interactions of elongating RNAPII, we endogenously fused UltraID to elongation factor ELOF1 using a knock-in approach^{5,7}. As a control, we knocked UltraID into the ELOF1 locus, flanked by a T2A peptide that causes ribosome skipping during translation and thereby produces two separate proteins, ELOF1 and UltraID, from a single mRNA transcript. **Indeed, this proximity-labelling approach showed that PAF1C interacts with elongating RNAPII in the absence of DNA damage (Fig 4F), as expected, while no interactions were detected with the UltraID only control.”*****

We fail to see how the goal of this experiment is not well described. The essential line to read are in bold. Suggestions for improving our phrasing are welcome.

The authors have gone to great lengths to do this by using an ELOF1-UltraID construct for biotin labeling of transiently associated proteins and they do show proximity of PAF1 and CTR9 to ELOF1. It is surprising this was not conducted in UV-treated cells given the focus on the manuscript.

Proximity labeling is very suitable to detect transient interactions since rapidly exchanging proteins will be labelled each time they associate with the complex, meaning a greater fraction of the pool is labelled. Stable interactions are poorly detected, since the stably bound proteins will only be labelled once. We have performed proximity labeling after UV, but we did not detect any significant UV-induced interactors because of this very reason. Classical IP's are suitable to detect stable interactions, but they are unsuitable to detect transient binders, such as elongation factors. This is why we performed both.

It is also surprising that RTF1 and Pol II were not tested. SPT5 is tested but not described sufficiently.

We can include RTF1 and explain SPT5. The problem with RNAPII is that it is very sticky and Co-IPs with the beads.

11. The Methods section lacks important detail on cell line construction. For DOT1L, the authors mention making a KO line, but this is not discussed in sufficiently. Was a homology cassette used? If so, what are the boundaries of the deletion?

Why would a homology cassette be used to generate a knock-out? This is used for a knock-in using homology-directed repair. Making knockouts using CRISPR-Cas9 is pretty standard and we describe it as follows:

“Parental RPE1-hTERT cells stably expressing inducible Cas9 (iCas9) that are also knockout for TP53 and the puromycin-N-acetyltransferase PAC1 gene were described previously (referred to as RPE1-iCas9). To generate knockouts in RPE1-iCas9, one day after induction of Cas9 by addition of 200 ng/ul doxycycline (Clontech, 8634-1), cells were transfected with 10nM crRNA:tracrRNA duplexes using 1:1000 Lipofectamine RNAiMAX transfection reagent (Invitrogen, 13778150). crRNAs in Supplementary Table 2. Two days after transfection, single cells were plated by limiting dilution. After single cell clone expansion, clones were selected by western blot and Sanger sequencing using the oligos listed in Supplementary Table 4.”

We can provide more information on the Sanger sequencing and the type of deletion.

For the other “depletions”, the authors again use CRISPR, but it is very unclear how these experiments work. Are the authors inducing a pool of mutants by inducing iCas9 and working with mixed populations?

Yes. The cells stably express inducible Cas9. We transfect cells with crRNAs after switching on Cas9 expression and we perform experiments on the pool. Since RNF20 and RPBB5 are essential, we can not generate stable knockouts.

In that case, are some cells mutant while others are wild type, so that the population as a whole has less of the target protein?

That is possible, which is why we measure histone mark levels and nascent transcript levels in individual cells, to assess how well the depletion worked per cell. This is also why performing ChIP-seq, as suggested by this reviewer, is not very useful. We note that the knock-out efficiency in the mixed-pool is very high, as evidence by our western blot experiments on the pool.

Adding to the confusion, the term “acutely depleted” misrepresents the long time-courses involved in inducing CRISPR (5 days). Sometimes the authors use the term “acute knockout”. Are these depletions or knockouts?

There is no misrepresentation. We knock out the gene by transfecting a crRNA in cells expressing Cas9. We then wait 5 days for the protein to be turned over and the histone marks to be turned-over as well. This is an acute knock-out of the gene, followed by depletion of the protein (since the gene encoding it has been knocked out). We can change it to acute knockout. We refer to them as acute since they are not stable knockouts as we use for DOT1L. Stable knockout of RNF20 and RPBB5 is not viable.

A more minor point: crRNA information for RBBP5 and RNF20 is lacking in the tables.

We will add this information.

12. The writing needs to be improved.

- a. There are six SET1 family members in human cells. On several occasions on page 5, the authors' description implies that there is only one. The text needs to be revised to clearly describe the complexity of this problem, and be more careful about using SET1 generically.
- b. The first heading in the Results states that "DOT1L and H3K79 are redundant for transcription restart". How can they be redundant when one is dependent on the other?
- c. Page 5. Line 39 citation to figure should be to Figure 2E and not 2D.
- d. Page 5. Line 58 citation to figure should be to Figure 3E and not 3D.
- e. Page 6. Header. Replace "is" with "are".
- f. RNAPII-S2 is an antibody to Ser2-phosphorylated Pol II CTD. This needs to be stated.
- g. EU is used in Figure 1 but not defined until Figure 2.
- h. NC needs to be defined when first used, especially in the figure legends.

We can address these textual suggestions.

Referee: 2

Transcription-coupled repair (TCR) of DNA damages that block transcription is an important strategy for cells to remove genotoxic lesions that could cause cell death. Following TCR, the mechanism(s) that mediate transcription restart have been partially elucidated in model species such as *C. elegans* and mouse, but information in human cells is lacking. The PAF1C complex has been implicated as an important factor in regulating transcriptional restart following DNA damage. However, the specific functions of PAF1C that contribute to transcription restart (in addition to PAF1) are not known and are the main subject of this study. In this manuscript, Luijsterburg and colleagues investigate requirements for DOT1L, H3K79 methylation, SET1 complex, H3K4 trimethylation, H2BK120-Ub, RNF20, Core PAF1 subunits PAF1, CTR9 and RTF1 in transcription restart. The experiments are well described and the conclusions drawn are in line with the observed results. With the exception of PAF1 and CTR9 (whose functions in restart have been previously reported), none of the many factors and histone modifications examined were found to function in transcription restart.

This study is mainly a collection of negative results for identifying which candidates might play a role in transcription restart following DNA damage repaired by TCR. Although some differences (i.e. DOT1L-mediated H3K79 methylation) between mouse and human cells are noted and several transcription-associated histone modifications can be ruled out, the findings reported do not substantially advance our understanding of the mechanism of transcription recovery after TCR of UV-induced DNA damage in human cells.

As the authors state in the final sentence of the manuscript, "It would be interesting to further dissect differences in species requirements of post-repair restart compared to those of general transcription." The authors should be encouraged to continue their investigations in order to identify the players involved in this process and take this work beyond simply eliminating candidates and processes that do not participate in restart.

We strongly disagree that negative results do not advance our understanding. It is important to show that transcription-associated histone marks are not required for transcription restart.

Referee: 3

The PAF1 elongation complex plays an intricate role in transcription. Following initiation, it wraps around the RNA Polymerase II complex to support transcription elongation. The PAF1 complex has several other roles. It is required for transcription restart after DNA damage and it promotes the deposition of histone modifications, particularly H3K4me3, H3K79 methylation, and H2B-K120ub. The RTF1 protein is a loosely associated subunit of the PAF1 complex. This special subunit plays a key role in transcription elongation and is responsible for promoting the downstream histone modifications. However, the functional significance of this for transcription restart remains unclear. Earlier studies demonstrated a role for DOT1L (H3K79 methylation) in transcription restart in mouse embryonic fibroblasts and in transcription recovery after induction of a transcription-replication conflict in human embryonic kidney cells.

The study by Van Schie et al addresses the role of the RTF1-dependent histone modifications in transcription restart after UV-induced damage in human near-diploid retinal pigment epithelial (RPE1) cells. In a concise and clear set of experiments the authors find that inactivation of the enzymes or enzyme complexes that deposit H3K4me3, H3K79 methylation, and H2B-K120ub (DOT1L, RBBP5, RNF20) does not affect the transcription restart 24h after DNA damage, while the respective modifications are undetectable or much reduced. Moreover, while knock-out of PAF1 complex subunits CTR9 and PAF1 compromises restart, knock-out of RTF1 had no effect, while it affected the downstream modifications more than knock-out of the other PAF1 complex subunits did. Finally, the authors find that the PAF1 complex but not RTF1 shows increased binding to the elongating form of RNA Pol II after UV damage. Together, these findings demonstrate that the stimulatory effect of the PAF1 complex, and particularly RTF1 on the three histone modifications is not required for the role of the PAF1 complex in transcription restart in human RPE1 cells.

Overall this concise paper is well written and structured, the literature is well cited, and most of the conclusions are supported by the data. The results are negative but interesting and important for the field.

Major comments:

1. In this paper, (recovery of) nascent RNA synthesis (RRS) by RNA Polymerase II is measured by incorporation of 5-EU, followed by click chemistry and imaging. Incorporation of 5-EU is not specific for RNA Pol II, 5-EU also labels RNA Pol I and III transcripts. The authors should explain in more detail the rationale of the assay, and how the global RRS assays used, without analysis of specific RNA molecules or other measure, directly relates to RNA Polymerase II or whether they might be biased by (abundant) RNA Pol I and III transcripts. Alternatively, additional experiments should be included to address this issue to specifically assess RNA Polymerase II transcripts, at least for the most important conditions.

RNAPIII transcript are too short to be affected by the dose of UV light we use here (leading to 1 CPD damage per 20 kb of transcribed DNA). RNAPI transcripts are made exclusively in nucleoli. The EU levels we measure here in the nucleoplasm are RNAPII transcripts.

Moreover, the PAF1 complex is a well-defined elongation factor for RNAPII.

We can re-analyze nascent transcription based on previously measured TT-seq after UV for RNAPI, RNAPII, and RNAPIII transcripts.

2. Related to this, the title 'PAF1C-driven restoration of RNAPII elongation after DNA damage..' seems too specific. In this paper, RNA Polymerase activity was studied by measuring global RNA synthesis. I suggest that 'RNAPII elongation' is replaced by 'transcription'.

We disagree. Both CSA and PAF1 are specific for RNAPII. As mentioned in the previous points, we can re-analyze TT-seq data to quantify RNAPI, RNAPII, and RNAPIII transcripts after UV.

3. The finding that DOT1L is not involved in transcription restart in the human RPE1 cells contrasts earlier findings mouse embryonic fibroblasts. The authors suggest that this might relate to species specific roles. However, another possibility is that it relates to the different cell types used. Overall, it is important that the authors explicitly and clearly mention in the results section and their conclusions that the studies were performed with human RPE1 cells (and explain the cell line in the results section). Confirmation in an independent human cell model (for example by DOT1L inhibition or knock-out of RTF1) would strengthen the claim that PAF1 complex/DOT1L might have species specific roles in transcription restart.

We can validate another cell-line using DOT1L inhibitor. We have previously shown that knock-down of PAF1 in U2OS cells affects transcription restart, so here we validate this in RPE1 cells (and extend it to another PAF1C subunit: CTR9).

4. While SET1 complexes and RNF20/40 predominantly specifically 'write' H3K4me3 and H2B-K120ub, respectively, DOT1L is responsible for H3K79 me1, me2 and me3. For most occasions where H3K79me2 is mentioned in the text, it would be more appropriate to refer to H3K79 methylation instead of H3K79me2. Otherwise the reader may get the wrong impression that specific functions have been ascribed to H3K79me2. Similarly, knock-out of DOT1L will lead to loss of all H3K79 methylation, not just loss of H3K79me2. The text and figure legends should be adjusted accordingly.

We will change this accordingly.

5. Fig 4 and Page 6. Line 18. To draw conclusions about how deletion of PAF1 complex subunits affects histone modifications, results of statistical tests should be shown in Fig. 4B. The legend should explain what the term 'western blots' refers to in terms of replicates (technical/biological).

We will change this and add the requested information.

6. The authors show in Fig 4F that Elongation factor ELOF1 is associated with CTR9 and PAF1 (using elegant endogenous tagging of ELOF1 with an UltraID tag) under normal conditions. The authors conclude that this confirmed that elongating RNA Polymerase II interacts with the PAF1 complex in the absence of DNA damage. The added value of this experiment for this paper is unclear to me since several lines of evidence have shown this interaction.

The RNAPII – PAF1 interaction has mainly been studied using in vitro reconstitution. It is quite difficult to detect binding of the PAF1 complex to RNAPII in cells. Our normal IP approach fails to detect this interaction due its transient nature. We are unaware of studies showing the RNAPII – PAF1 interaction using IP in human cells.

Moreover, even though UltraID cannot be applied directly to RNA Pol II S2-phospho, studying the 'proximal interactome' of ELOF1 is not a direct measure of the interactome of the elongating form of RNA Polymerase II. Additional experiments will be needed to confirm the findings and to determine if these interactions indeed occur on the DNA template. The authors should give more context and/or perform additional experiments to confirm that the ELOF1 interactome is a validated proxy for the elongating RNA Pol II interactome.

We perform proximity labelling using elongation factor ELOF1. As a control, we knock-in UltraID into the ELOF1 locus (without fusing it to ELOF1). We show that PAF1, CTR9, as well as elongation factor SPT5 are in close proximity to ELOF1 (but these proteins are not labelled by UltraID only). Moreover, we detect PAF1, CTR9 and SPT5 in close proximity to ELOF1 in CSB-KO cells, showing that PAF1C binding to the elongation complex does not depend on CSB. We have performed mass spec on ELOF1-UltraID and we detect many transcription elongation factors, showing this is a good proxy for elongating RNAPII. However, this dataset adds very little to the current manuscript and we intend to publish it separately. We could stain for more elongation factors, but that would be besides the point. The point we wish to make is that PAF1 can be detected this way, and that its association to elongating RNAPII does not require CSB. In contrast, after UV in a classical IP, we detect stable PAF1 binding to RNAPII in a manner that is fully dependent on CSB. We could add a cartoon to clarify this better.

Other points:

7. Page 3. Line 21. The abbreviation NER has not been explained yet.

This has been changed.

8. Page 4. Line 49, and elsewhere. The authors cannot conclude that the modification loss is 'complete' or 'near complete'. Western blots are usually not very sensitive. Undetectable would be a more accurate description. Although the authors acknowledge this later in the text, such claims are better avoided. Mass spectrometry would be a more sensitive method but this also has detection limits.

We agree and will change the wording. Note that we also confirm this by immunostaining, not just western blotting.

9. Page 5. Line 19 and beyond. The authors use inducible Cas9 and transfection to knock-out essential genes. They use the term 'acutely depleted'. This term is usually used for rapid protein degradation strategies using degrons or protacs. In the scenario of iCas9, the term 'conditionally depleted' would be more intuitive. The conditional knock-out protocol for studying essential genes is not described in the methods.

We agree to using conditional knock-out. We will provide a full protocol in the methods.

10. Page 58. Line 58. The authors conclude that: 'H2BK120ub is dispensable.' However, this statement is too strong because H2B-K120ub is only partially reduced on blots. The results do show that wild-type levels of H2BK120ub are dispensable.

Agree and we will change this accordingly.

11. Supplementary Table 3: Plasmid pML#355 (...ultaID..) refers to Addgene plasmid #7284. This Addgene plasmid is a auxin inducible degron plasmid. Is this correct?

Yes, this is the backbone that was used. We can provide full plasmid maps.

Dear Martijn,

Thank you for the submission of your manuscript with referee reports to EMBO reports. I have read through all files now and think that your study will make a nice contribution to our scientific reports section.

I would like to ask you to address all referee concerns as you suggest and will send the revised ms only back to referee 3. I will ask *[journal name redacted]* for referee 3's identity.

Please address all referee concerns in a complete point-by-point response. Acceptance of the manuscript will depend on a positive outcome of a second round of review. It is EMBO reports policy to allow a single round of major revision only and acceptance or rejection of the manuscript will therefore depend on the completeness of your responses included in the next, final version of the manuscript.

In the interest of protecting the conceptual advance provided by the work, we recommend a revision within 3 months (21st Nov 2025). In your case, the revised ms can probably be submitted earlier.

You can either publish the study as a short report or as a full article. For short reports, the revised manuscript should not exceed 29,000 characters (including spaces but excluding materials & methods and references) and 5 main plus 5 expanded view figures. The results and discussion sections must further be combined, which will help to shorten the manuscript text by eliminating some redundancy that is inevitable when discussing the same experiments twice. For a normal article there are no length limitations, but it should have more than 5 main figures and the results and discussion sections must be separate. In both cases, the entire materials and methods must be included in the main manuscript file.

- 1) A data availability section providing access to data deposited in public databases is missing. If you have not deposited any data, please add a sentence to the data availability section that explains that.
- 2) Your manuscript contains statistics and error bars based on $n=2$. Please use scatter blots in these cases. No statistics should be calculated if $n=2$.

5) a complete author checklist, which you can download from our author guidelines . Please insert information in the checklist that is also reflected in the manuscript. The completed author checklist will also be part of the RPF.

6) Please note that all corresponding authors are required to supply an ORCID ID for their name upon submission of a revised manuscript (). Please find instructions on how to link your ORCID ID to your account in our manuscript tracking system in our Author guidelines

- the name of the statistical test used to generate error bars and P values,
- the number (n) of independent experiments (please specify technical or biological replicates) underlying each data point,
- the nature of the bars and error bars (s.d., s.e.m.),
- If the data are obtained from n {less than or equal to} 2, use scatter blots showing the individual data points.

12) All Materials and Methods need to be described in the main text using our 'Structured Methods' format, which is required for all research articles. According to this format, the Methods section includes a separate Reagents and Tools Table file (listing key reagents, experimental models, software and relevant equipment and including their sources and relevant identifiers) and a Methods and Protocols section describing the methods using a step-by-step protocol format. The aim is to facilitate adoption of the methodologies across labs. More information on how to adhere to this format as well as a downloadable template (.docx) for the Reagents and Tools Table can be found in our author guidelines:

An example of a Method paper with Structured Methods can be found here: <https://www.embopress.org/doi/full/10.1038/s44320-024-00037-6#sec-4>

You are able to opt out of this by letting the editorial office know (emboreports@embo.org). If you do opt out, the Review Process File link will point to the following statement: "No Review Process File is available with this article, as the authors have

chosen not to make the review process public in this case."

I look forward to seeing a revised form of your manuscript when it is ready.

Referee: 1

In this manuscript, van Schie et al. follow up on prior work from the Luijsterburg group (van den Heuvel et al., Nat. Commun. 2021), which showed a requirement for PAF1 in restarting transcription after DNA damage in U2OS cells. In the current manuscript, the authors investigate if deposition of PAF1-dependent histone marks is required for transcription restart post-UV irradiation in RPE1-hTERT cells. Using CRISPR-based strategies to induce mutations or knock out genes for DOT1L, RBBP5, and RNF20, the authors generate cell lines to test the roles in transcription restart for H3K79Me2, H3K4Me3, and H2BK120Ub, respectively. They also generate cell lines to deplete members of the PAF complex: PAF1, RTF1, and CTR9. Transcription restart after UV irradiation is assayed by EU-labeling of cells, and histone modifications are measured in the same cells by immunofluorescence. A final set of experiments tests association of PAF subunits with Pol II and elongation factor ELOF1.

The primary new conclusion from this manuscript is that transcription restart after DNA damage appears to occur normally in the absence of PAF-dependent histone modifications, and therefore, some other function of PAF1, which remains undefined, is required to restore transcription elongation after repair of DNA damage. In general, the data are convincing, although limited in depth, and they support the argument that no single PAF-dependent histone mark can explain PAF1's role in transcription restart. The authors should address the following comments and also a discrepancy between their current data and published work.

Specific comments:

1. Statistics are lacking for the graphs.

Statistics have been added.

2. Figure 1. Unlike the other histone marks, H3K79Me2 levels in the individual cells are not shown. This information should be provided. Also, why do the authors focus on the dimethylation state?

Unlike the other marks, DOT1L^{KO} in RPE1 cells is viable and therefore stable KO clones can be generated in which H3K79me is fully lost. It is therefore unnecessary to measure H3K79me levels in individual cells. We show by western blot that the mark is lost. Also, the commercial antibody we use for H3K79me2 does not work properly for immunofluorescence. We do not focus on dimethylation state per se, but use the H3K79me2 antibody as a method to detect H3K79 methylation status and have clarified this in the text.

3. Figure 1d. Nascent transcription appears to be decreased in nonirradiated DOT1L-KO cells. A comment is needed.

We added a comment in the text.

4. Figure 2A. A blot showing depletion of RBBP5 is needed. In addition, the levels of H3K4Me2 should be tested, especially since previous work has shown different effects of H3K4Me2 and H3K4Me3 on DNA repair.

We show loss of H3K4me₃ by western blot and by immunostaining. Showing depletion of RBBP5 will add nothing new to this, as we already show that the histone mark is gone. The transcription-associated mark linked to PAF1 function is H3K4me₃, not H3K4me₂. Moreover, both are fully dependent on RBBP5.

5. Figure 2D. The authors state that H3K4Me3 intensity is not changed in the CSA crRNA cells. This conclusion does not reflect the data at the 24-hour timepoint, where H3K4Me3 levels seem to be lower than for the NC-treated cells. Applying the < 0.2% cutoff seems quite stringent.

Indeed, H3K4me₃ levels are slightly but significantly lower at 24 h after UV in crCSA conditions. We now comment on this in the text. The cut-off is 0.2 in decimal form, meaning less than 20% of the intensity in crNC cells. We see how this annotation might be confusing and have changed it.

6. Figure 3A. Depletion of RNF20 should be confirmed by western blot.

Showing RNF20 depletion by western blot adds no information. We measure H2B_{Ub} by western blotting and immunostaining at the single cell level, where we correlate this to nascent transcript levels (measured by EU labelling), showing that RNF20 is functionally depleted.

7. Figure 3D. In the van den Heuvel et al. paper, H2BK120Ub was shown to recover on gene bodies 8 hours post-UV treatment. In Figure 3D, H2BK120Ub levels are not recovered after 24 hours in the NC crRNA-treated cells. Because H2BK120Ub is a mark of active transcription, it is surprising to see recovery of transcription without recovery of this mark. Their previous observations seem more likely.

We performed new imaging experiments up to 48h. Following a pronounced decrease at 3 hours, parental cells exhibited full transcription recovery at 24 and 48 hours, accompanied by a corresponding drop and subsequent restoration of H2BK120_{Ub} levels. At 24 hours, H2BK120_{Ub} levels had not yet returned to pre-UV levels, and increased slightly further at 48 hours (Fig 4A-C). In CSB^{KO} cells, transcription failed to recover at both 24 and 48 hours, and H2BK120_{Ub} levels remained as low as at 3 hours post-UV (Fig 4A-C), suggesting that H2BK120_{Ub} levels track transcription recovery, potentially with some delay.

The authors should provide an orthologous measure of H2BK120Ub in their crRNA-treated cells, such as ChIP-seq (as in van den Heuvel et al.) and also explain the discrepancies in their results.

To orthogonally validate these findings, we performed genome-wide ChIP-seq for H2BK120_{Ub} and reanalyzed our previous BrU-seq dataset, which measures genome-wide nascent RNA synthesis (van der Weegen et al., 2021). BrU-seq in RPE1 cells revealed a strong decrease in nascent transcription within gene bodies 3 hours after UV, consistent with photolesion frequency and the likelihood of RNAPII encountering them (Perdiz et al., 2000). This effect is evident in heatmaps of 3,000 genes sorted by length (up to 100 kb), a genome browser track for the GALNT1 gene, and a metagene analysis of 919 genes ≥50 kb or 390 genes ≥100 kb (Fig 4D-F). H2BK120_{Ub} followed a similar trend, although decreases were primarily detected beyond 20-30 kb, whereas nascent transcription was already sharply reduced beyond 10 kb (Fig 4D-F). This suggests that H2B is not immediately deubiquitylated when nascent transcription is lost. Indeed, H2BK120_{Ub} levels dropped to ~40% at 50 kb and ~20% at 100 kb into gene bodies, whereas nascent transcription had already fallen to background levels at these positions (Fig 4D-F). At 24 hours post-UV, both nascent transcription and H2BK120_{Ub} levels fully recovered even up to 100 kb. Thus, although H2BK120_{Ub} is not required for transcription recovery after UV, the levels of this co-transcriptional mark correlate with

transcription recovery, indicating that it is deposited behind RNAPII without being necessary for the restart itself.

8. The authors have switched cell lines from their earlier study. An explanation for choice of cell line is needed.

We previously used U2OS cells, which are aneuploid cancer cells with high transcription levels and highly variable transcription recovery. We switched to RPE1 cells, which are near-diploid with transcription levels similar to primary fibroblasts and much more reproducible transcription recovery between experiments.

9. Figure 4E. This is essentially a repeat of Figure 2 in the van den Heuvel et al. paper. The only significant difference is showing RTF1 is not associated with Pol II after UV treatment. In our previous paper we used U2OS cells, this experiment is done in RPE1 cells. It is not a repeat, but the result between U2OS and RPE1 cells is indeed the same: CSB is essential to recruit the PAF1 complex subunits PAF1 and CTR9. In addition, we show that RTF1 is not recruited after UV irradiation.

10. Figure 4F. The goal of this experiment is not well-described, but I believe the authors are primarily testing if PAF subunits CTR9 and PAF1 can be detected in the elongation complex under normal conditions, which is something that has been well documented in many experiments.

The primary goal of this experiment is to show the PAF1C in undamaged conditions binds to RNAPII independently of CSB (now Fig 6F), in contrast to the stable interaction between PAF1C and RNAPII after UV which is dependent on CSB (Fig 6D). These experiments show these are two different binding modes which likely represent the different functions of PAF1C in elongation versus transcription restart. We have improved the description of these experiments.

The authors have gone to great lengths to do this by using an ELOF1-UltraID construct for biotin labeling of transiently associated proteins and they do show proximity of PAF1 and CTR9 to ELOF1. It is surprising this was not conducted in UV-treated cells given the focus on the manuscript.

Proximity labeling is very suitable to detect transient interactions since rapidly exchanging proteins will be labelled each time they associate with the complex, meaning a greater fraction of the pool is labelled. Stable interactions are poorly detected, since the stably bound proteins will only be labelled once. We have performed proximity labeling after UV, but we did not detect any significant UV-induced interactors because of this very reason. Classical IP's are suitable to detect stable interactions, but they are unsuitable to detect transient binders, such as elongation factors. This is why we performed both. We have added a cartoon to clarify the different methods.

It is also surprising that RTF1 and Pol II were not tested. SPT5 is tested but not described sufficiently.

We have included RTF1 in Fig 6F. We stained for SPT5 as a control to show we detect general elongation factors using this assay. We have prepared a new extended view figure extending the validation of our ELOF1-ultraID approach showing that RNAPII itself as well as several

other known transcription elongation factors, including SPT5, SPT6, IWS1, CTR9, and RTF1, were also detected (Fig. 6F).

11. The Methods section lacks important detail on cell line construction. For DOT1L, the authors mention making a KO line, but this is not discussed in sufficiently. Was a homology cassette used? If so, what are the boundaries of the deletion?

No homology cassette was used. We generated knockouts through a standard CRISPR/Cas9-based approach, which we describe as follows:

“Parental RPE1-hTERT cells stably expressing inducible Cas9 (iCas9) that are also knockout for TP53 and the puromycin-N-acetyltransferase PAC1 gene were described previously (referred to as RPE1-iCas9). To generate knockouts in RPE1-iCas9, one day after induction of Cas9 by addition of 200 ng/ul doxycycline (Clontech, 8634-1), cells were transfected with 10nM crRNA:tracrRNA duplexes using 1:1000 Lipofectamine RNAiMAX transfection reagent (Invitrogen, 13778150). crRNAs in Supplementary Table 2. Two days after transfection, single cells were plated by limiting dilution. After single cell clone expansion, clones were selected by western blot and Sanger sequencing using the oligos listed in Supplementary Table 4.”

DOT1L^{KO}s were generated using two different crRNA targeting different regions in DOT1L. The sequences of the used crRNAs are provided in the methods sections. We have verified by sequencing that DOT1L^{KO}s have out of frame indels.

For the other “depletions”, the authors again use CRISPR, but it is very unclear how these experiments work. Are the authors inducing a pool of mutants by inducing iCas9 and working with mixed populations?

Yes. The cells stably express inducible Cas9. We transfect cells with crRNAs after switching on Cas9 expression and we perform experiments on the pool. Since RNF20 and RPBB5 are essential, we can not generate stable knockouts. To improve clarity, we added a separate description in the methods section for the generation of conditional knockouts.

In that case, are some cells mutant while others are wild type, so that the population as a whole has less of the target protein?

That is possible, which is why we measure histone mark levels and nascent transcript levels in individual cells (for H2BK120_{Ub} and H3K4me₃), to assess how well the depletion worked per cell. We note that the knock-out efficiency in the mixed-pool is high, as evidenced by our western blot experiments on the pool, and IF for single cells.

Adding to the confusion, the term “acutely depleted” misrepresents the long time-courses involved in inducing CRISPR (5 days). Sometimes the authors use the term “acute knockout”. Are these depletions or knockouts?

This is an acute knock-out of the gene, followed by depletion of the protein (since the gene encoding it has been knocked out). We have changed the term to ‘conditional knockout’ to avoid confusion. Stable knockout of RNF20 and RPBB5 is not viable.

A more minor point: crRNA information for RBBP5 and RNF20 is lacking in the tables. Information has been added.

12. The writing needs to be improved.

a. There are six SET1 family members in human cells. On several occasions on page 5, the authors' description implies that there is only one. The text needs to be revised to clearly describe the complexity of this problem, and be more careful about using SET1 generically.

b. The first heading in the Results states that "DOT1L and H3K79 are redundant for transcription restart". How can they be redundant when one is dependent on the other?

c. Page 5. Line 39 citation to figure should be to Figure 2E and not 2D.

d. Page 5. Line 58 citation to figure should be to Figure 3E and not 3D.

e. Page 6. Header. Replace "is" with "are".

f. RNAPII-S2 is an antibody to Ser2-phosphorylated Pol II CTD. This needs to be stated.

g. EU is used in Figure 1 but not defined until Figure 2.

h. NC needs to be defined when first used, especially in the figure legends.

We thank the reviewer for these textual suggestions, which have all been incorporated.

Referee: 2

Transcription-coupled repair (TCR) of DNA damages that block transcription is an important strategy for cells to remove genotoxic lesions that could cause cell death. Following TCR, the mechanism(s) that mediate transcription restart have been partially elucidated in model species such as *C. elegans* and mouse, but information in human cells is lacking. The PAF1C complex has been implicated as an important factor in regulating transcriptional restart following DNA damage. However, the specific functions of PAF1C that contribute to transcription restart (in addition to PAF1) are not known and are the main subject of this study. In this manuscript, Luijsterburg and colleagues investigate requirements for DOT1L, H3K79 methylation, SET1 complex, H3K4 trimethylation, H2BK120-Ub, RNF20, Core PAF1 subunits PAF1, CTR9 and RTF1 in transcription restart. The experiments are well described and the conclusions drawn are in line with the observed results. With the exception of PAF1 and CTR9 (whose functions in restart have been previously reported), none of the many factors and histone modifications examined were found to function in transcription restart.

This study is mainly a collection of negative results for identifying which candidates might play a role in transcription restart following DNA damage repaired by TCR. Although some differences (i.e. DOT1L-mediated H3K79 methylation) between mouse and human cells are noted and several transcription-associated histone modifications can be ruled out, the findings reported do not substantially advance our understanding of the mechanism of transcription recovery after TCR of UV-induced DNA damage in human cells.

As the authors state in the final sentence of the manuscript, "It would be interesting to further dissect differences in species requirements of post-repair restart compared to those of general transcription." The authors should be encouraged to continue their investigations in order to identify the players involved in this process and take this work beyond simply eliminating candidates and processes that do not participate in restart.

We disagree that negative results do not advance our understanding. It is important to show that transcription-associated histone marks are not required for transcription restart.

Referee: 3

The PAF1 elongation complex plays an intricate role in transcription. Following initiation, it wraps around the RNA Polymerase II complex to support transcription elongation. The PAF1 complex has several other roles. It is required for transcription restart after DNA damage and it promotes the deposition of histone modifications, particularly H3K4me3, H3K79 methylation, and H2B-K120ub. The RTF1 protein is a loosely associated subunit of the PAF1 complex. This special subunit plays a key role in transcription elongation and is responsible for promoting the downstream histone modifications. However, the functional significance of this for transcription restart remains unclear. Earlier studies demonstrated a role for DOT1L (H3K79 methylation) in transcription restart in mouse embryonic fibroblasts and in transcription recovery after induction of a transcription-replication conflict in human embryonic kidney cells.

The study by Van Schie et al addresses the role of the RTF1-dependent histone modifications in transcription restart after UV-induced damage in human near-diploid retinal pigment epithelial (RPE1) cells. In a concise and clear set of experiments the authors find that inactivation of the enzymes or enzyme complexes that deposit H3K4me3, H3K79 methylation, and H2B-K120ub (DOT1L, RBBP5, RNF20) does not affect the transcription restart 24h after DNA damage, while the respective modifications are undetectable or much reduced. Moreover, while knock-out of PAF1 complex subunits CTR9 and PAF1 compromises restart, knock-out of RTF1 had no effect, while it affected the downstream modifications more than knock-out of the other PAF1 complex subunits did. Finally, the authors find that the PAF1 complex but not RTF1 shows increased binding to the elongating form of RNA Pol II after UV damage. Together, these findings demonstrate that the stimulatory effect of the PAF1 complex, and particularly RTF1 on the three histone modifications is not required for the role of the PAF1 complex in transcription restart in human RPE1 cells.

Overall this concise paper is well written and structured, the literature is well cited, and most of the conclusions are supported by the data. The results are negative but interesting and important for the field.

Major comments:

1. In this paper, (recovery of) nascent RNA synthesis (RRS) by RNA Polymerase II is measured by incorporation of 5-EU, followed by click chemistry and imaging. Incorporation of 5-EU is not specific for RNA Pol II, 5-EU also labels RNA Pol I and III transcripts. The authors should explain in more detail the rationale of the assay, and how the global RRS assays used, without analysis of specific RNA molecules or other measure, directly relates to RNA Polymerase II or whether they might be biased by (abundant) RNA Pol I and III transcripts. Alternatively, additional experiments should be included to address this issue to specifically assess RNA Polymerase II transcripts, at least for the most important conditions.

Please note that 5-EU incorporation measures levels of nascent transcription. The RNAPI and RNAPIII transcript are abundant because they are stable, not because they are transcribed at much higher levels than RNAPII transcripts. Moreover, we note that the PAF1 complex is a well-defined elongation factor for RNAPII.

Nonetheless, to quantify the relative contribution of the three RNA polymerases, we measured 5-EU levels following treatment of cells with inhibitors of either RNAPI, RNAPII, or, RNAPIII, showing RNAPII is the main contributor (about 75% of total EU) of nascent transcription in our experiments. These new results are shown in Figure EV1.

2. Related to this, the title 'PAF1C-driven restoration of RNAPII elongation after DNA damage..' seems too specific. In this paper, RNA Polymerase activity was studied by measuring global RNA synthesis. I suggest that 'RNAPII elongation' is replaced by 'transcription'.

We changed the title to: "PAF1C restores transcription after DNA damage independently of histone mark deposition."

3. The finding that DOT1L is not involved in transcription restart in the human RPE1 cells contrasts earlier findings mouse embryonic fibroblasts. The authors suggest that this might relate to species specific roles. However, another possibility is that it relates to the different cell types used. Overall, it is important that the authors explicitly and clearly mention in the results section and their conclusions that the studies were performed with human RPE1 cells (and explain the cell line in the results section). Confirmation in an independent human cell model (for example by DOT1L inhibition or knock-out of RTF1) would strengthen the claim that PAF1 complex/DOT1L might have species specific roles in transcription restart.

We confirmed that transcription restart does not require DOT1L in human fibroblasts (48BR-hTERT), using DOT1L inhibitor (shown in Figure 1D-F). We have previously shown that knock-down of PAF1 in U2OS cells affects transcription restart, so here we validate this in RPE1 cells. We have now also extended this finding to another PAF1C subunit: CTR9 (shown in Figure 5C, D).

4. While SET1 complexes and RNF20/40 predominantly specifically 'write' H3K4me3 and H2B-K120ub, respectively, DOT1L is responsible for H3K79 me1, me2 and me3. For most occasions where H3K79me2 is mentioned in the text, it would be more appropriate to refer to H3K79 methylation instead of H3K79me2. Otherwise the reader may get the wrong impression that specific functions have been ascribed to H3K79me2. Similarly, knock-out of DOT1L will lead to loss of all H3K79 methylation, not just loss of H3K79me2. The text and figure legends should be adjusted accordingly.

We have changed this accordingly.

5. Fig 4 and Page 6. Line 18. To draw conclusions about how deletion of PAF1 complex subunits affects histone modifications, results of statistical tests should be shown in Fig. 4B. The legend should explain what the term 'western blots' refers to in terms of replicates (technical/biological).

We have changed this accordingly and have added the requested information.

6. The authors show in Fig 4F that Elongation factor ELOF1 is associated with CTR9 and PAF1 (using elegant endogenous tagging of ELOF1 with an UltraID tag) under normal conditions. The authors conclude that this confirmed that elongating RNA Polymerase II interacts with the PAF1 complex in the absence of DNA damage. The added value of this experiment for this paper is unclear to me since several lines of evidence have shown this interaction.

Indeed it has been documented that PAF1C interacts with RNAPII, which has been almost exclusively studied using in vitro reconstitution with recombinant proteins. Our IP approaches fail to detect this interaction due to the transient nature of the RNAPII-PAF1C interaction in cells.

The primary goal of this experiment is to show that the binding of PAF1C to RNAPII is independent of CSB, and thus is different from the UV-induced stable RNAPII-PAF1C interaction which is CSB-dependent. Thereby, these experiments show these are two different binding modes which are likely central to the different functions of PAF1C in elongation versus transcription restart. We have clarified our intentions with this experiment in the text and added a cartoon to clarify the IP versus the UltraID approach.

Moreover, even though UltraID cannot be applied directly to RNA Pol II S2-phospho, studying the 'proximal interactome' of ELOF1 is not a direct measure of the interactome of the elongating form of RNA Polymerase II. Additional experiments will be needed to confirm the findings and to determine if these interactions indeed occur on the DNA template. The authors should give more context and/or perform additional experiments to confirm that the ELOF1 interactome is a validated proxy for the elongating RNA Pol II interactome.

We have provided an additional supplemental figure (shown in Figure EV2) validating that our ELOF1-UltraID approach specifically detects elongation factors. We show that RNAPII-S2, STP5, SPT6, IWS1, PPP1CB, CTR9 and PAF1 are in close proximity to ELOF1. As a control, we knocked-in UltraID into the ELOF1 locus without resulting in a protein fusion to ELOF1, the unfused ultraID indeed does not detect elongation complex members. Moreover, we detect PAF1, CTR9 and SPT5 in close proximity to ELOF1 in CSB-KO cells, showing that PAF1C binding to the elongation complex does not depend on CSB. The point we wish to make is that PAF1 can be detected this way, and that its association to elongating RNAPII does not require CSB. In contrast, after UV in a classical IP, we detect stable PAF1 binding to RNAPII in a manner that is fully dependent on CSB. We have added a cartoon to clarify this better.

Other points:

7. Page 3. Line 21. The abbreviation NER has not been explained yet.

This has been changed.

8. Page 4. Line 49, and elsewhere. The authors cannot conclude that the modification loss is 'complete' or 'near complete'. Western blots are usually not very sensitive. Undetectable would be a more accurate description. Although the authors acknowledge this later in the text, such claims are better avoided. Mass spectrometry would be a more sensitive method but this also has detection limits.

We agree and have changed the wording. Note that we also confirm this by immunostaining, not just western blotting.

9. Page 5. Line 19 and beyond. The authors use inducible Cas9 and transfection to knock-out essential genes. They use the term 'acutely depleted'. This term is usually used for rapid protein degradation strategies using degrons or protacs. In the scenario of iCas9, the term

'conditionally depleted' would be more intuitive. The conditional knock-out protocol for studying essential genes is not described in the methods.

To avoid confusion, we agree to using conditional knock-out. We will provide a full protocol in the methods.

10. Page 58. Line 58. The authors conclude that: 'H2BK120ub is dispensable.' However, this statement is too strong because H2B-K120ub is only partially reduced on blots. The results do show that wild-type levels of H2BK120ub are dispensable.

We agree and have changed this accordingly.

11. Supplementary Table 3: Plasmid pML#355 (...ultaID..) refers to Addgene plasmid #7284. This Addgene plasmid is a auxin inducible degron plasmid. Is this correct?

Yes, this is the backbone that was used.

Dear Martijn,

Thank you for the submission of your revised manuscript. We have now received the enclosed report from referee 1 (previously referee 3) who was asked to assess it. This referee has some more suggestions that I would like you to address and incorporate before we can proceed with the official acceptance of your manuscript.

A few editorial requests will also need to be addressed:

- The "Disclosure and Competing Interests Statement" subheading needs to be corrected.
- The author credits need to be removed from the ms file. All credits need to be entered during online ms submission.
- The REFERENCE format is not correct; et al needs to be used after 10 author names. Please correct to the EMBO reports reference format.
- In the author CHECKLIST all responses to the pull-down menu in column D are missing. Please submit a fully completed checklist with your final ms.
- There are 6 suppl. tables in the ms that should all be part of the Reagents & Tools table file instead. The Materials and Methods section should include a separate Reagents and Tools Table file (listing key reagents, experimental models, software and relevant equipment and including their sources and relevant identifiers) and a Methods and Protocols section in which methods should be described using a step-by-step protocol format with bullet points. More information is available in our guide to authors online: <https://link.springer.com/journal/44319/submission-guidelines>
- The manuscript sections should be in the following order: Title page - Abstract & Keywords - Introduction - Results - Discussion - Methods - Data Availability - Acknowledgments - Disclosure Statement & Competing Interests - References - Figure Legends - (Main Tables with legends if applicable) - Expanded View Figure Legends.
- The nomenclature of EV figure legends needs correction to Figure EV1, Figure EV2 (Figure Expanded View 1, etc. is not OK).
- In the Data Availability Section, the specific URL that directly leads to the deposited dataset needs to be added.

* Figure Legends - Comments *

- Please note that the exact p values are not provided in the legends of figures 1C, F, H, J; 2D, E; 3D, E; 4B, C; 5B, D; EV1 B, please provide exact values as reasonable.
- Please indicate the statistical test used for data analysis in the legends of figures 6A, B
- Please note that information related to n is missing in the legends of figures 1H, 2C-E; 3C-E; 4B, C; 6A, B; EV1 B
- Please note that the scale bar needs to be defined for figures 1B, E
- Please note that the dashed borders are not defined in the legend of figure 1B, E, I; 2B, 3B, 5C. This needs to be rectified.

EMBO press papers are accompanied online by A) a short (1-2 sentences) summary of the findings and their significance, B) 2-3 bullet points highlighting key results and C) a synopsis image that is exactly 550 pixels wide and 200-600 pixels high (the height is variable). The synopsis image should provide a sketch of the major findings, like a graphical abstract. Please note that text needs to be readable at the final size. Please send us this information along with the final manuscript.

Referee #1:

Based on the comments of the reviewers on a previous version of this manuscript, Van Schie et al performed additional experiments and made several textual changes. The new results and clarifications have further strengthened the paper.

The PAF1 complex is known to play a role in transcription restart after DNA damage and to promote deposition of three transcription-associated histone modifications. These three histone modifications have also been implicated in DNA damage response and transcription restart after DNA damage. However, their precise involvement, especially in human cells, remained unclear. The results of Van Schie et al show that the three histone modifications are dispensable for transcription restart after DNA damage in human cells, as is RTF1, the loosely associated subunit of the PAF1 complex that is primarily responsible for enhancing deposition of the three histone modifications. Thereby, this study suggests that the PAF1 complex in human cells must affect transcription restart by other mechanisms.

The authors have addressed most of the comments of the reviewers. However, several issues still need to be resolved.

1. To make the title more specific and avoid confusion, the authors could change the second part of the title to: independently of promoting histone mark deposition.
2. The rationale for performing the ELOF1-UltraID experiments (Fig 6 and page 8-9) is now more clearly explained. However, the description of the results and the conclusions still require some adjustments. In particular, the authors should describe the comparison between RNAPII-S2 co-IP and ELOF1-UltraID with more care and nuance. There are two issues that need rewriting. First, interactions with the elongating form of RNAPII and interactions with one of the elongation factors cannot be directly compared. Differences between these experiments may at least in part be attributed to the different baits used. The authors should at least discuss this possibility. In describing the UltraID results, the authors should not refer to interactions with elongating RNAPIII but instead refer to interactions with ELOF1. They can subsequently discuss that ELOF1 interactions likely reflect interactions with elongating RNAPII. Second, the authors suggest that proximity labeling is very suitable for transient interactions and that stable interactions are poorly detected with this method. They refer to Kubitz et al. This reviewer could not find evidence for this statement in Kubitz et al, nor in other publications and notes that proximity labeling has been successfully used to identify stable interactions. Therefore, the authors should provide more evidence for this statement or remove the cartoon panels C and E in Fig. 6 and adjust the writing in the results section, e.g. explain that UltraID allows detection of stable and transient interactions, and refrain from specifically referring to transient interactions later in the text.
3. When applying the statistical tests, the authors should consider the notion that in many of the figures, the mock sample consists of normalized values where all values equal 1 (most of the figures that show quantification of the RRS assays). Therefore, these values do not represent a normal distribution. In such cases, a test to determine if the median of the normalized values is significantly different from 1 seems appropriate. In addition, the way the results are normalized is not consistent, e.g. Fig. 2E and 3E are different from the other panels. If data are normalized within a condition, e.g. NC, RNF20, CSA, the authors should plot these as individual plots or at least add separator lines to avoid direct comparisons between conditions (e.g. Fig 2D should be three separated parts, while combining all samples into one plot seems fine for 2E).
4. In the same RRS quantification plots, the authors highlight some samples as non-significant (ns) and others as significant (*) while many are not labeled. The use of this is not consistent and not explained. To avoid a bias in interpreting the results, the authors should provide the same information, ns or significant, for all the relevant comparisons in all the figures.

Minor issues:

1. Page 7, Fig. 4F: the authors should justify the grouping of genes in 'above 50kb' and 'above 100 kb'. To avoid confusion, I suggest referring to the first group as 'above 50 kb and below 100 kb', if this is indeed the correct description.
2. The authors should justify the choice of the cell lines used in the main text. In addition, following comment 9 of Reviewer 1, the authors should explain in the text how Fig. 4E relates to previously published work in U2OS cells.
3. Page 5. The authors cannot conclude that no obvious H3K79 demethylase exists. It would be more accurate to state that no obvious H3K79 demethylase has been identified. Here the authors could cite a more recent review such PMID: 35733849.
4. Page 8, end of first paragraph: Since similar results have previously been observed in other cell lines, please change show to confirm: these results confirm that these subunits.
5. Legends: Please explain UT in the legend of Fig 1A and NC in the legend of Fig 2A.

EMBOR-2025-62555V2

Editorial requests

(1) The "Disclosure and Competing Interests Statement" subheading needs to be corrected.

Done.

(2) The author credits need to be removed from the ms file. All credits need to be entered during online ms submission.

Done.

(3) The REFERENCE format is not correct; et al needs to be used after 10 author names. Please correct to the EMBO reports reference format.

Done.

(4) In the author CHECKLIST all responses to the pull-down menu in column D are missing. Please submit a fully completed checklist with your final ms.

Done.

(5) There are 6 suppl. tables in the ms that should all be part of the Reagents & Tools table file instead. The Materials and Methods section should include a separate Reagents and Tools Table file (listing key reagents, experimental models, software and relevant equipment and including their sources and relevant identifiers) and a Methods and Protocols section in which methods should be described using a step-by-step protocol format with bullet points. More information is available in our guide to authors online: <https://link.springer.com/journal/44319/submission-guidelines>

We put the supplemental tables in the Reagents & Tools table format.

(6) The manuscript sections should be in the following order: Title page - Abstract & Keywords - Introduction - Results - Discussion - Methods - Data Availability - Acknowledgments - Disclosure Statement & Competing Interests - References - Figure Legends - (Main Tables with legends if applicable) - Expanded View Figure Legends.

Done.

(7) The nomenclature of EV figure legends needs correction to Figure EV1, Figure EV2 (Figure Expanded View 1, etc. is not OK).

Done.

(8) In the Data Availability Section, the specific URL that directly leads to the deposited dataset needs to be added.

Done

Figure Legends - Comments

(9) Please note that the exact p values are not provided in the legends of figures 1C, F, H, J; 2D, E; 3D, E; 4B, C; 5B, D; EV1 B, please provide exact values as reasonable.

We added the exact p values directly to the figures.

(10) Please indicate the statistical test used for data analysis in the legends of figures 6A, B

Done.

(11) Please note that information related to n is missing in the legends of figures 1H, 2C-E; 3C-E; 4B, C; 6A, B; EV1 B

Information is added.

(12) Please note that the scale bar needs to be defined for figures 1B, E

Done.

(13) Please note that the dashed borders are not defined in the legend of figure 1B, E, I; 2B, 3B, 5C. This needs to be rectified.

Dashed borders are now defined.

(14) EMBO press papers are accompanied online by A) a short (1-2 sentences) summary of the findings and their significance, B) 2-3 bullet points highlighting key results and C) a synopsis image that is exactly 550 pixels wide and 200-600 pixels high (the height is variable). The synopsis image should provide a sketch of the major findings, like a graphical abstract. Please note that text needs to be readable at the final size.

Summary:

PAF1C contributes to transcription restart following DNA damage, independently of RTF1 and promoting histone mark deposition.

Bullet points:

- H3K79_{me}, H3K4_{me3}, and H2BK120_{Ub} deposition is dispensable for transcription restart after DNA damage.
- PAF1C contributes to transcription restart after DNA damage independent of dissociable subunit RTF1.
- CSB specifically stabilizes PAF1C on RNAPII in response to DNA damage, but not during processive transcription elongation.

Referee #1:

Based on the comments of the reviewers on a previous version of this manuscript, Van Schie et al performed additional experiments and made several textual changes. The new results and clarifications have further strengthened the paper.

The PAF1 complex is known to play a role in transcription restart after DNA damage and to promote deposition of three transcription-associated histone modifications. These three histone modifications have also been implicated in DNA damage response and transcription restart after DNA damage. However, their precise involvement, especially in human cells, remained unclear. The results of Van Schie et al show that the three histone modifications are dispensable for transcription restart after DNA damage in human cells, as is RTF1, the loosely associated subunit of the PAF1 complex that is primarily responsible for enhancing deposition of the three histone modifications. Thereby, this study suggests that the PAF1 complex in human cells must affect transcription restart by other mechanisms.

The authors have addressed most of the comments of the reviewers. However, several issues still need to be resolved.

(1) To make the title more specific and avoid confusion, the authors could change the second part of the title to: independently of promoting histone mark deposition.

We have changed the title accordingly.

(2) The rationale for performing the ELOF1-UltraID experiments (Fig 6 and page 8-9) is now more clearly explained. However, the description of the results and the conclusions still require some adjustments. In particular, the authors should describe the comparison between RNAPII-S2 co-IP and ELOF1-UltraID with more care and nuance. There are two issues that need rewriting. First, interactions with the elongating form of RNAPII and interactions with one of the elongation factors cannot be directly compared. Differences between these experiments may at least in part be attributed to the different baits used. The authors should at least discuss this possibility.

We changed the wording to further clarify our approach.

In describing the UltraID results, the authors should not refer to interactions with elongating RNAPIII but instead refer to interactions with ELOF1. They can subsequently discuss that ELOF1 interactions likely reflect interactions with elongating RNAPII.

Proximity labelling does not capture interactions with ELOF1. It captures proteins that were in close proximity to ELOF1 (within 10-20 nm = 100-200 Å). Note that RNAPII is about 100–120 Å across the entire complex. Therefore, it would be factually incorrect to refer to identified proteins as interactors of ELOF1, as proteins could be located on the either side of RNAPII and still be labelled. We changed the wording to further clarify our approach.

Second, the authors suggest that proximity labeling is very suitable for transient interactions and that stable interactions are poorly detected with this method. They refer to Kubitz et al. This reviewer could not find evidence for this statement in Kubitz et al, nor in other publications and notes that proximity labeling has been successfully used to identify stable interactions. Therefore, the authors should provide more evidence for this statement or remove the cartoon panels C and E in Fig. 6 and adjust the writing in the results section, e.g. explain that UltraID allows detection of stable and transient interactions, and refrain from specifically referring to transient interactions later in the text.

We never mention that stable interactions are poorly detected by proximity labelling. We do mention that proximity labelling is more suitable to detect transient interactions than co-IP. This is evident from many studies, including Kubitz et al., but also Göös et al. 2022 (PMID: 35140242) and González-

Vinceiro et al. 2025 (PMID: 41202128). We now cite both studies. Stable interactions will be labelled less than transient interactions, because more stably bound proteins exchange less and are therefore labelled less. This is inherent to the principle of proximity labelling.

Göös et al. mention that: *“It has been suggested that the BioID method is efficient for studying transient interactions, and this was supported by our results.”*

A review by Varnaitè and MacNeill (PMID: 27329485) argue similarly: *“BioID offers significant advantages over conventional interactome discovery methods, particularly in regards to the identification of transient or weak interactions.”*

(3) When applying the statistical tests, the authors should consider the notion that in many of the figures, the mock sample consists of normalized values where all values equal 1 (most of the figures that show quantification of the RRS assays). Therefore, these values do not represent a normal distribution. In such cases, a test to determine if the median of the normalized values is significantly different from 1 seems appropriate.

Where conditions are compared to values normalized to 1, we used a one sample t-test with hypothetical value 1, we have further clarified this in the figure legends. In the case of the RRS assays the relevant comparison is the 24h timepoint of crNC/WT with the 24h timepoint of the crRNA/KO condition (which are normalized to their respective mock (no UV) condition, and thus are not value 1 and are normally distributed, in which case we performed one-way ANOVA's, which we describe in the figure legends.

In addition, the way the results are normalized is not consistent, e.g. Fig. 2E and 3E are different from the other panels. If data are normalized within a condition, e.g. NC, RNF20, CSA, the authors should plot these as individual plots or at least add separator lines to avoid direct comparisons between conditions (e.g. Fig 2D should be three separated parts, while combining all samples into one plot seems fine for 2E).

All RRS experiments are consistently normalized to their respective mock (no UV) condition; this normalization and the used way of plotting the data is common practise for RRS experiments after UV (see: PMID: 41554717, 41535461, 39547229, 38316879, 36816995, 39547223, 39021334, 38600236, and many more).

When H3K4_{me3} or H2B_{Ub} levels are quantified (2E and 3E) or when mock EU incorporation is compared (e.g. Fig. 1E, 2C), all levels are normalized to mock non-targeting control (NC) or WT, since here the absolute differences matter between conditions (for instance crRBBP5 will lower H3K4_{me3} levels, so normalizing to crRBBP5 mock does not make sense)

We describe in the figure legends to which condition we normalize. In addition, since all values of the normalized condition are exactly 1, we find that normalization is also directly visually apparent from the graphs.

(4) In the same RRS quantification plots, the authors highlight some samples as non-significant (ns) and others as significant (*) while many are not labeled. The use of this is not consistent and not explained. To avoid a bias in interpreting the results, the authors should provide the same information, ns or significant, for all the relevant comparisons in all the figures.

To improve clarity, we added p values directly to the figures. All relevant comparisons are labelled with a p value. Where comparisons are not relevant, p values are not determined and therefore these are not labelled with a p value.

Minor issues:

(5) Page 7, Fig. 4F: the authors should justify the grouping of genes in 'above 50kb' and 'above 100 kb'. To avoid confusion, I suggest referring to the first group as 'above 50 kb and below 100 kb', if this is indeed the correct description.

It is common practise to look at longer genes, since used doses of UV irradiation trigger about 1 CPD every ~20 kb in the transcribed strand. This means that the probability of having a damage in an active gene increases as genes are longer. We have therefore plotted genes of at least 50 kb and genes of at least 100 kb.

The suggested description 'above 50 kb and below 100 kb' is incorrect, the group 'above 50 kb', includes all genes above 50kb, and the group 'above 100 kb', includes all genes above 100 kb. We changed the annotation to ' ≥ 50 kb' and ' ≥ 100 kb' to avoid any confusion.

(6) The authors should justify the choice of the cell lines used in the main text. In addition, following comment 9 of Reviewer 1, the authors should explain in the text how Fig. 4E relates to previously published work in U2OS cells.

RPE1 are commonly used, human non-transformed diploid cells, and we have therefore used them in this study and multiple studies before this one, we thus see no need to justify this choice. We added explanation to relate our results to the previously published work in U2OS cells.

(7) Page 5. The authors cannot conclude that no obvious H3K79 demethylase exists. It would be more accurate to state that no obvious H3K79 demethylase has been identified. Here the authors could cite a more recent review such PMID: 35733849.

We changed the phrasing and added the citation.

(8) Page 8, end of first paragraph: Since similar results have previously been observed in other cell lines, please change show to confirm: these results confirm that these subunits.

We have changed this as requested.

(9) Legends: Please explain UT in the legend of Fig 1A sand NC in the legend of Fig 2A.

We added the explanations.

Prof. Martijn Luijsterburg
Leiden University Medical Center
Department of Human Genetics
Leiden 2333 ZC
Netherlands

Dear Martijn,

I am very pleased to accept your manuscript for publication in the next available issue of EMBO reports. Thank you for your contribution to our journal.

You may qualify for financial assistance for your publication charges - either via a Springer Nature fully open access agreement or an EMBO initiative. Check your eligibility: <https://link.springer.com/journal/44319/how-to-publish-with-us>

>>> Please note that it is EMBO Reports policy for the transcript of the editorial process (containing referee reports and your response letter) to be published as an online supplement to each paper. If you do NOT want this, you will need to inform the Editorial Office via email immediately. More information is available here: <https://link.springer.com/partners/embo-press/editorial-policies#Peer%20review>